# Spiking neurons with spatiotemporal dynamics and gain modulation for monolithically integrated memristive neural networks

Qingxi Duan[1], Zhaokun Jing[1], Xiaolong Zou[4], Yanghao Wang [1], Ke Yang[1], Teng Zhang[1], Si Wu[4], Ru Huang [1,2,3✉] & Yuchao Yang [1,2,3✉]

As a key building block of biological cortex, neurons are powerful information processing units and can achieve highly complex nonlinear computations even in individual cells. Hardware implementation of artificial neurons with similar capability is of great significance for the construction of intelligent, neuromorphic systems. Here, we demonstrate an artificial neuron based on $NbO_x$ volatile memristor that not only realizes traditional all-or-nothing, threshold-driven spiking and spatiotemporal integration, but also enables dynamic logic including XOR function that is not linearly separable and multiplicative gain modulation among different dendritic inputs, therefore surpassing neuronal functions described by a simple point neuron model. A monolithically integrated 4 × 4 fully memristive neural network consisting of volatile $NbO_x$ memristor based neurons and nonvolatile $TaO_x$ memristor based synapses in a single crossbar array is experimentally demonstrated, showing capability in pattern recognition through online learning using a simplified δ-rule and coincidence detection, which paves the way for bio-inspired intelligent systems.

[1] Key Laboratory of Microelectronic Devices and Circuits (MOE), Department of Micro/nanoelectronics, Peking University, Beijing 100871, China. [2] Center for Brain Inspired Chips, Institute for Artificial Intelligence, Peking University, Beijing 100871, China. [3] Frontiers Science Center for Nano-optoelectronics, Peking University, Beijing 100871, China. [4] School of Electronics Engineering and Computer Science, Peking University, Beijing 100871, China. ✉email: ruhuang@pku.edu.cn; yuchaoyang@pku.edu.cn

The human brain consists of extremely dense networks of computational (neurons) and memory elements (synapses), all of which operate at very low energy levels, using only 20 fJ per operation[1,2]. There are ~$10^{11}$ neurons in the human brain, with the corresponding ~$10^{15}$ total synapses[1,3]. The neuron is composed of different parts, where soma is the main body of neurons and are connected to neural networks via dendrites and axons. The dendrites are responsible for receiving information (inputs) from other neurons, while the axons are responsible for transmitting the information out (output)[4]. When a neuron is activated, it releases a signal (pulse) that is transmitted down the axon and through the synapse into the dendrites of the next neuron. The weight of the synapse, that is, the strength of the connection between neurons, can be made stronger or weaker by a process called synaptic plasticity when the brain adapts to new information. Inspired by the structure and principles of the human brain, neuromorphic computing has great potential in the next generation of computing technology, with massive parallelism and high efficiency, hence holding great prospect in overcoming the bottleneck of von Neumann architecture and extending the boundary of intelligence. To achieve this goal, the development of highly compact artificial neurons and synapses, especially that capture important neuronal and synaptic dynamics, as well as the construction of hardware systems based on such artificial elements are of great significance.

Traditional complementary metal-oxide semiconductor (CMOS) technology utilizes complex auxiliary circuits and bulky capacitors to simulate bio-dynamics, and is not suitable for constructing scalable neurons and synapses due to area and energy inefficiencies[5]. In recent years, many researchers have devoted extensive efforts into the development of artificial synapses using emerging nonvolatile memory devices with abundant ion dynamics[6–8], such as nonvolatile memristors[9–11], diffusive memristors[12,13], synaptic transistors[14,15], etc. In contrast, although artificial neurons are of equal importance, the study on compact, highly functional neuron is relatively limited. A large number of recent theoretical works have revealed that a

single neuron can be used as a highly powerful unit of computation[16–18], whose functionality far exceeds a simple point neuron model. Many of the important functionalities, such as gain modulation, have not been implemented in artificial neurons to date, including Hodgkin-Huxley neurons[19,20], leaky integrate and fire (LIF) neurons[21–23], and oscillation neurons[24,25], etc. Beyond the exploration for more capable neuromorphic devices, there are very few reports on integration of artificial synaptic devices with artificial neurons in the same array and hardware construction of a fully memristive neural network for functional demonstrations[26,27].

In this study, we report an artificial neuron based on $NbO_x$ volatile memristor that not only realizes all-or-nothing, threshold-driven spiking and spatiotemporal integration, but also enables dynamic logic and gain modulation among different dendritic inputs, therefore going beyond the functions of a simple point neuron model. Such artificial neuron with spatiotemporal dynamics and gain modulation could form an integral part for biologically plausible, fully memristive neural networks, where both the neurons and synapses are built by compact memristive devices. As an example, a monolithically integrated $4 \times 4$ spiking neural network consisting of volatile $NbO_x$ memristor-based neurons and nonvolatile $TaO_x$ memristor-based synapses in a single crossbar array is experimentally demonstrated, showing capability in performing pattern recognition through online learning based on a simplified δ-rule and achieving coincidence detection. Meanwhile, a 3-layer spiking neural network based on $NbO_x$ neuron and $TaO_x$ synapse is further constructed by simulation, achieving an accuracy of 83.24% on MNIST handwritten digit classification. The multiplicative gain modulation of the artificial neuron has been utilized to achieve receptive field remapping, which can potentially enhance the stability of artificial visual systems.

## Results

**Artificial neurons based on $NbO_x$ volatile memristors**. A schematic diagram of a simply connected biological neural

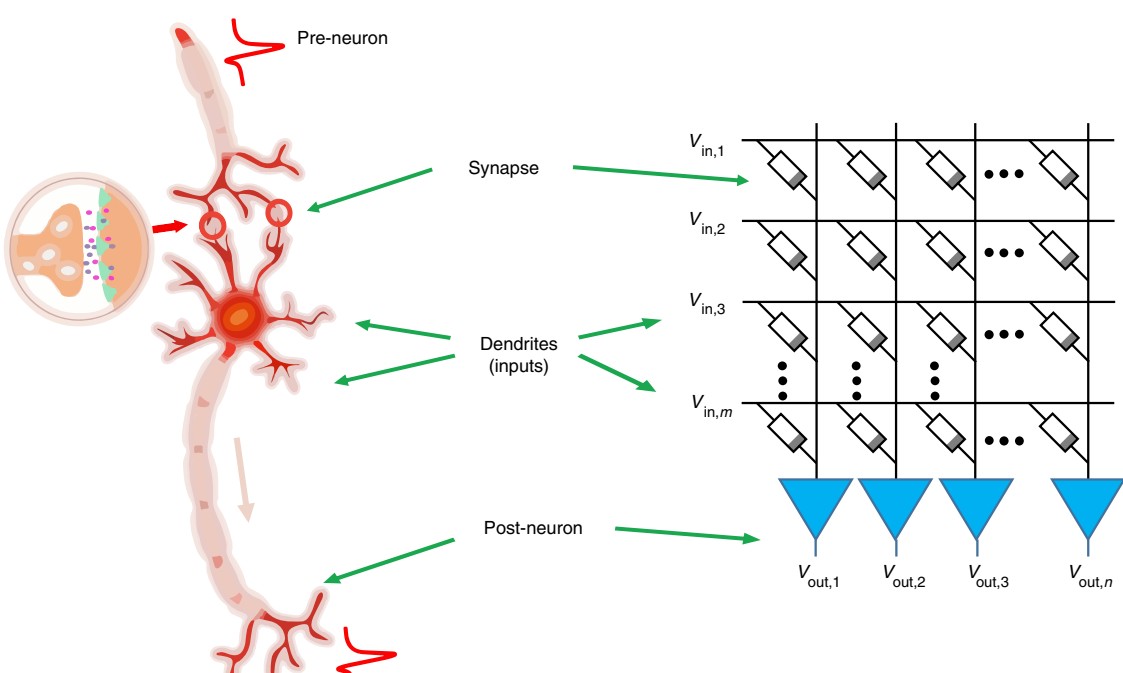

**Fig. 1 Comparison of biological and artificial neurons.** Schematic of biological neurons and synapses (left) compared with an artificial neural network (right).

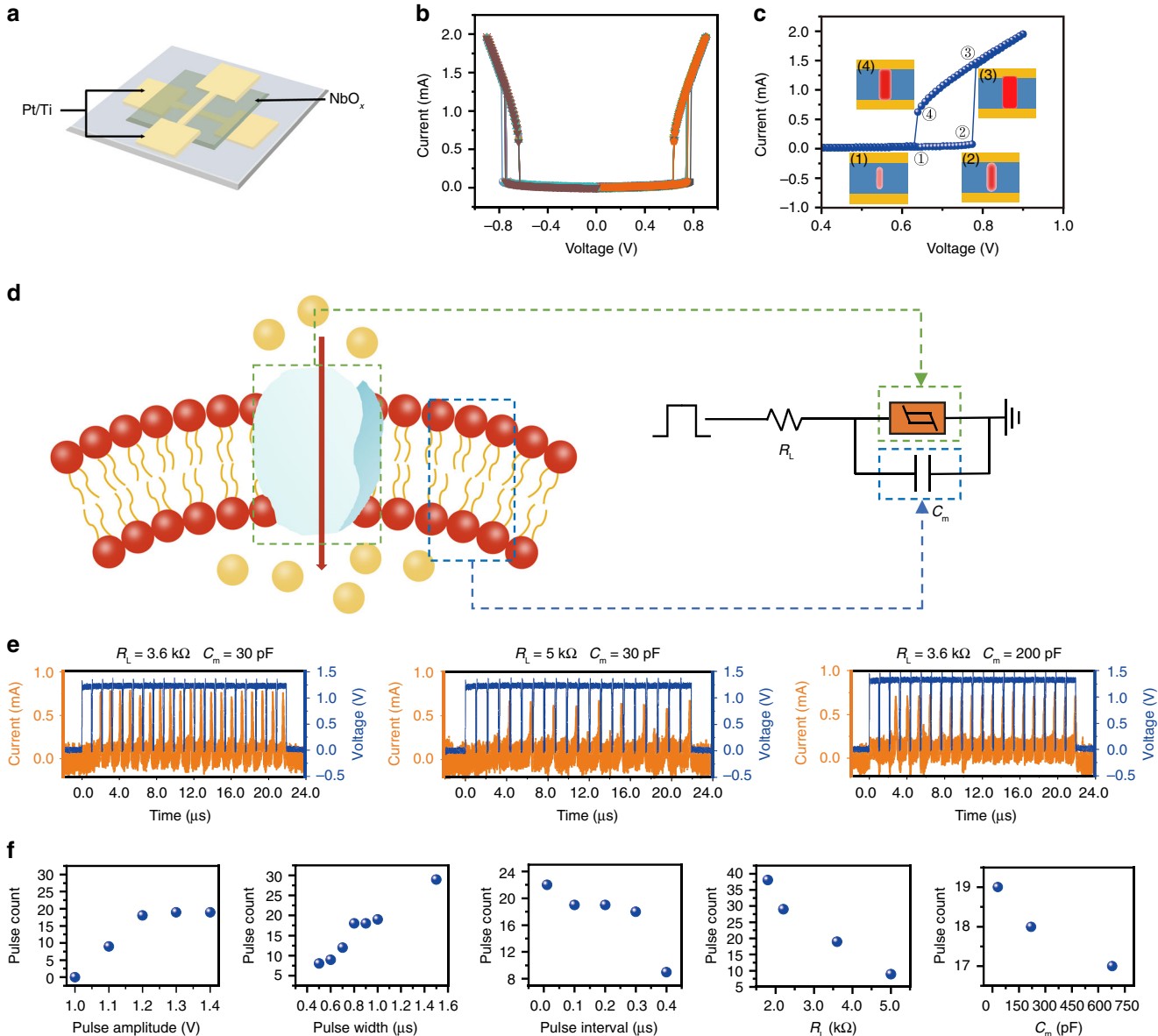

**Fig. 2 Spiking neuron based on NbO$_x$ volatile memristors. a** Schematic diagram of the memristive device, which consists of a NbO$_x$ layer sandwiched between two Pt/Ti electrodes. **b** Current-voltage characteristics of the device repeated for 10 cycles. **c** Schematic illustration of the mechanism responsible for threshold switching in the NbO$_x$ devices. **d** Illustration of an ion channel embedded in cell membrane of a biological neuron, with corresponding circuit based on NbO$_x$ devices for implementation of a spiking neuron. **e** Characterization of the LIF neuron under a continuous pulse train and the influence of varying capacitance ($C_m$) and resistance ($R_L$). **f** The firing response of the spiking neuron under different input and circuit conditions. When changing the input parameters such as the pulse width, pulse interval and circuit parameters such as $C_m$ and $R_L$, the firing pulse count has an obvious change with 20 input pulse cycles.

network is shown in Fig. 1. The input signals received through the dendrites are integrated by the soma of the neuron, which in turn generates action potentials and are then transmitted by the axon. When the neurons are not exposed to external stimuli, the electrical balance of neurons is maintained by various ions, such as Na$^+$, K$^+$, Cl$^-$, and Ca$^{2+}$, moving in and out of the neuronal cells[28]. The potential difference between the interior and exterior of the neuron is called the membrane potential. When they are exposed to external stimuli, the membrane potential of an excitatory neuron increases due to Na$^+$ influx, which triggers an action potential when the membrane potential exceeds a certain threshold. Such behavior of biological neurons is described in an abstract form, i.e., by the leaky integrate and fire (LIF) model[29], which can be emulated by volatile memristors herein, by

sandwiching a NbO$_x$ film between two Pt/Ti electrodes, as schematically illustrated in Fig. 2a (see Supplementary Fig. 1 for scanning electron microscopy (SEM) and transmission electron microscopy (TEM) results of the Pt/Ti/NbO$_x$/Pt/Ti device, also see Supplementary Note 1). The device can be switched to high conductance state when the applied voltage exceeds a threshold ($V_{th}$) of ~±0.8 V without an external current compliance and spontaneously returns to the low conductance state once the applied voltage drops below a holding voltage ($V_{hold}$) of ~±0.6 V (Fig. 2b). Symmetric hysteresis loops were observed in both bias polarities (Fig. 2b), and the device can operate properly after >10$^9$ switching cycles (Supplementary Fig. 2, Supplementary Note 1), showing stable volatile threshold switching (TS) characteristics. Transient electrical measurements show that the switching speed

of NbO$_x$ threshold switching device in the present work is <50 ns from off- to on-state and <25 ns from on- to off-state (Supplementary Fig. 3, Supplementary Note 1), with acceptable cycle-to-cycle and device-to-device variations, making them qualified for building artificial neurons (Supplementary Figs. 4 and 5, Supplementary Note 1). Such threshold switching characteristics in NbO$_x$-based memristors have attracted extensive attention[30–34], and recent studies revealed that such effects can be well interpreted by a trap-assisted conduction mechanism similar to Poole-Frenkel model with moderate Joule heating, therefore suggesting an electronic nature and much lower switching temperature than the previous insulator-metal transition model[30,31]. Figure 2c schematically depicts the dynamic evolution of the device state during the threshold switching process: when the voltage exceeds ~±0.8 V, the device switches from off to on state due to the formation and expansion of a filament (state (1) to (3)), and the size of the filament decreases gradually from state (3) to state (4) when the applied voltage decreases. Finally, the filament does not have sufficient power to sustain itself, and the device returns back to off state (state (4) to (1)).

The LIF neuron can therefore be implemented by the simple circuit in Fig. 2d, where the Pt/Ti/NbO$_x$/Pt/Ti volatile memristor is connected in series with a load resistor ($R_L$) and a capacitor ($C_m$) is connected in parallel. $C_m$ could be either external capacitor or intrinsic capacitance of the Pt/Ti/NbO$_x$/Pt/Ti device. The TS behavior of the Pt/Ti/NbO$_x$/Pt/Ti device can mimic the dynamics of an ion channel located near the soma of a neuron, while the membrane capacitance is represented by $C_m$. Figure 2e shows the spiking behavior of the Pt/Ti/NbO$_x$/Pt/Ti-based artificial neuron as a function of detailed circuit parameters, where a pulse train with pulse width of 1 μs and amplitude of 1.3 V was applied. The interval between adjacent pulses is set to 0.1 μs. Due to the voltage dividing effect, the $C_m$ will be charged initially, since the voltage mainly drops across the Pt/Ti/NbO$_x$/Pt/Ti device ($R_{OFF} > R_L$, where $R_{OFF}$ is the initial resistance of NbO$_x$ threshold switching device). Once the voltage of the capacitor reaches $V_{th}$, the TS device will switch from off to on state, that is, the artificial LIF neuron fires, as manifested by a current spike (Fig. 2e). After the device switches to on state, the voltage across $R_L$ and the TS device will be re-distributed, and the capacitor begins to discharge. When the voltage drops below $V_{hold}$, the TS device returns to off state (Fig. 2e). Supplementary Fig. 6 further shows the input voltage, current response and the voltage across the threshold switching device as functions of time, which illustrate the dynamic switching process clearly. The firing rate, however, strongly depends on $R_L$ and $C_m$, as shown in Fig. 2e, f. While a smaller $C_m$ makes the integration process faster, a larger $R_L$ reduces the input current and slows down the charge build-up, hence delaying or preventing the firing, as shown in Fig. 2e, f. In addition, the characteristics of the LIF neuron can also be regulated by adjusting pulse parameters. The spiking frequency increases with the increased input stimulus in biological neurons[35]. To emulate this behavior, a series of pulse trains with varied pulse amplitude (1, 1.1, 1.2, 1.3, 1.4 V) and pulse width (0.5, 0.6, 0.7, 0.8, 0.9, 1.0, 1.5 μs) were applied to the artificial neuron, showing that the spiking frequency also increases as the pulse amplitude or pulse width increases, due to the overall increased charging speed, showing that the artificial neuron can successfully realize the strength-modulated spike frequency characteristics of biological neurons (Fig. 2f). Moreover, the spiking frequency decreases as the pulse interval increases due to leakage of the charge through the Pt/Ti/NbO$_x$/Pt/Ti device in parallel with $C_m$, thus implementing the leaky dynamics in a LIF neuron. Supplementary Fig. 7 further exhibits a plot of spike count vs. input frequency (with $R_L$ of 3.6 kΩ, $C_m$ of 30 pF, pulse width of 1 μs, and pulse amplitude of 1.2 V), where the pulse count increases as the input frequency increases and get saturated at very high frequency (see further discussion in Supplementary Note 2). Such rich tuning ability allows us to tailor the properties of the artificial neuron to achieve desirable characteristics for specific applications.

**Spatial and spatiotemporal integrations.** A fundamental computation that neurons perform is the transformation of incoming synaptic information into a specific spike pattern at output[17]. An important step of this transformation is neuronal integration, including the addition of events occurring simultaneously through different synapses (spatial summation) and addition of non-simultaneous events (temporal summation)[36,37]. Emulating this function in artificial neurons has a vital significance for computation and memory in neuromorphic hardware. Therefore, we use the circuit diagram in Fig. 3a to emulate this function, where the load resistors ($R_1$ and $R_2$) in series with NbO$_x$ threshold switching device serve as synapses distributed in space and a capacitor ($C_m$) is connected in parallel. Figure 3b is a schematic diagram of the integration of two presynaptic inputs with simultaneous stimuli (0.9 V in amplitude, 1 μs in width, 0.1 μs in interval, repeated for 10 cycles), where the two presynaptic inputs are integrated at the postsynaptic neuron. Detailed parameters for the synaptic weights and $C_m$ of the postsynaptic neuron are summarized in Supplementary Table 1. One can see that when only one input ($V_1$ or $V_2$) is applied, the neuron can perform temporal summation and fire a spike after a few pulses are applied. While the firing frequency of the neuron is ~0.36 MHz if only one input is applied to $S_1$ or $S_2$, the firing frequency is significantly increased to ~1.6 MHz when both presynaptic inputs are applied simultaneously, as shown in Fig. 3c, therefore achieving spatial summation. Systematic measurements by varying the amplitudes of both presynaptic inputs (from 0.6 to 0.9 V, with the pulse width, frequency and number fixed) show that the spatial summation can be extended to various input strengths, as plotted in Fig. 3d (see Supplementary Note 3 for fitting of the experimental data). Similarly, spatial summation can also be observed when the input pulse frequency is varied with the pulse amplitude, width and number fixed, as shown in Fig. 3f–h, where the collective effect of applying both presynaptic inputs results in higher firing frequency at the postsynaptic neuron than the case of single input (Fig. 3h, see Supplementary Note 3 for fitting of the experimental data).

Moreover, spatiotemporal summation of input information can also be achieved by the NbO$_x$ neuron. As shown in Fig. 3j, two time-dependent pulse trains (0.9 V in amplitude, 1 μs in width, 0.1 μs in interval, repeated for 10 cycles) were applied on $S_1$ and $S_2$, respectively, and the firing frequency of the artificial neuron is also a function of the time interval between the two pulse trains. As illustrated in Fig. 3k, when $S_1$ is applied before $S_2$ with a time difference of 2.2 μs ($\Delta t = 2.2$ μs), the firing spike count of the neuron is 8 within the same time duration (11 μs). When $S_1$ and $S_2$ are applied in synchronization, the firing spike count of the neuron is 11. When $S_1$ is applied after $S_2$ with a time difference of 2.2 μs ($\Delta t = -2.2$ μs), the firing spike count of neuron is 7. Figure 3l illustrates a spatiotemporal summation established by the spatiotemporally correlated spikes. The characteristics of the spatiotemporal summation can be generalized by a Lorentz function:

$$y = y_0 + A \times \left( \frac{B}{4 \times (\Delta t - \tau)^2 + B^2} \right) \quad (1)$$

where $y_0 = 6.04$, $A = 10.34$, $B = 2.15$, $\tau = 0.05$ for the fitting curve presented in Fig. 3l. As a result, presynaptic spikes from different synapses can trigger a postsynaptic spike in the postsynaptic

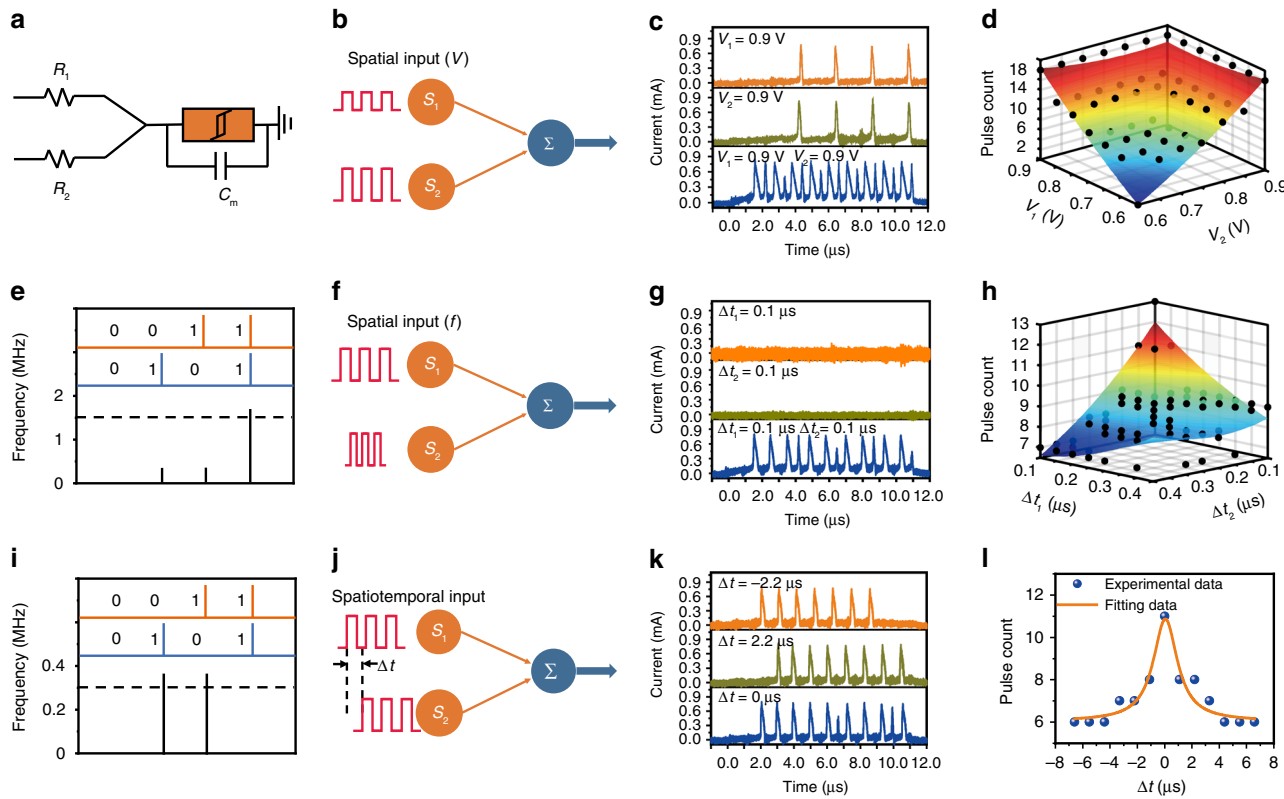

**Fig. 3 Spatiotemporal integration and dynamic logic in NbOₓ based neuron. a** The circuit diagram of the spiking neuron. **b** Schematic diagram of spatial summation with different input pulse amplitudes. **c** Neuronal response triggered by two input pulses (0.9 V amplitude, 1 μs width, interval 0.1 μs, 10 pulse cycles) applied on $S_1$ and $S_2$ individually and simultaneously. **d** The spatial summation at varied conditions and corresponding fitting results. The amplitudes of the two input spikes ($V_1$ and $V_2$) were systematically changed from 0.6 to 0.9 V, and applied simultaneously to the respective nodes. **e** The input–output characteristics for "AND" logic with two input pulses applied on $S_1$ and $S_2$. **f** Schematic diagram of spatial summation with different input pulse intervals. **g** Neuronal response triggered by two input pulses (0.8 V amplitude, 1 μs width, interval 0.1 μs, 10 pulse cycles) applied on $S_1$ and $S_2$ individually and simultaneously. **h** The spatial summation at varied conditions and corresponding fitting results. The intervals of the two input spikes were systematically changed from 0.05 to 0.4 μs. **i** The input–output characteristics for "XOR" logic with two input pulses applied on $S_1$ and $S_2$. **j** Schematic diagram of spatiotemporal summation with different time intervals of input pulses. **k** Neuronal response triggered by two input pulses (0.9 V amplitude, 1 μs width, 0.1 μs interval) applied on $S_1$ and $S_2$ with different time intervals of 0, 2.2 and −2.2 μs. **l** Spatiotemporal summation results when the time interval of two input spikes is changed from −6.6 to 6.6 μs. The solid line is fitting result by Eq. (1).

artificial neuron to establish spatiotemporal dynamics, similar with that in biological neural networks[38,39]. Such an artificial neuron is essential for spike-timing-dependent logic and neural computation.

Based upon the above results, we can also realize Boolean logic functions using the above artificial neuron (Fig. 3e). To do this, the input voltages to $S_1$ or $S_2$ serve as input variables, with 0 V and 0.9 V defined as logic "0" and "1", respectively, while the firing frequency of the artificial neuron is taken as the logic output, whose threshold is set to 1.5 MHz. As a result, only when the input signals are both 1, the firing frequency of the neuron exceeds the threshold, as shown in Fig. 3e, which experimentally implements the AND logic. Similarly, OR function can be implemented by defining a different threshold. More importantly, the spiking neuron in the present study can also be utilized to implement XOR function, which is a typical Boolean logic that is not linearly separable and therefore cannot be realized by a linear neural network[40]. To do this, the neuronal threshold is set to 0.3 MHz, and the logic "1" for $S_1$ or $S_2$ is designated to be 0.9 V and −0.9 V, respectively. Given the fact that the threshold switching in NbOₓ is independent on the input voltage polarity (Fig. 2b, Supplementary Figs. 4 and 5), the firing frequency of the neuron exceeds the threshold when either $p$ or $q$ is "1" but cannot reach the threshold when $p = 1$ and $q = 1$ are applied simultaneously, as

experimentally demonstrated in Fig. 3i and Supplementary Fig. 8. The successful implementation of logic functions that are not linearly separable, like XOR, further demonstrates the potential of the artificial neuron in achieving complex computing.

**Neuronal gain modulation in single artificial neuron.** The huge computing power of the brain has traditionally been thought of as being produced by complex neural network connections, where a single neuron acts as a simple linear summation and thresholding device. However, recent studies have shown that a single neuron has more complex nonlinear operation to convert synaptic inputs into output signals. These nonlinear mechanisms enable neurons to perform a series of arithmetic operations, which provide simple neurons with considerable computing power[17], and implementation of artificial neurons with such enhanced capability is of great significance for building real neuromorphic systems. Here we show that a spiking neuron can achieve neuronal modulation. Two types of presynaptic inputs are correlated to such neural transmission: one is the driving input ($V_d$), which enables the relevant neurons to fire strongly; the other is the modulatory input ($V_m$), which tunes the effectiveness of the driving input[16–18], as schematically illustrated in Fig. 4a. Silver[17] had reported the output of neurons under modulatory input and driving input and further suggested that neurons have the

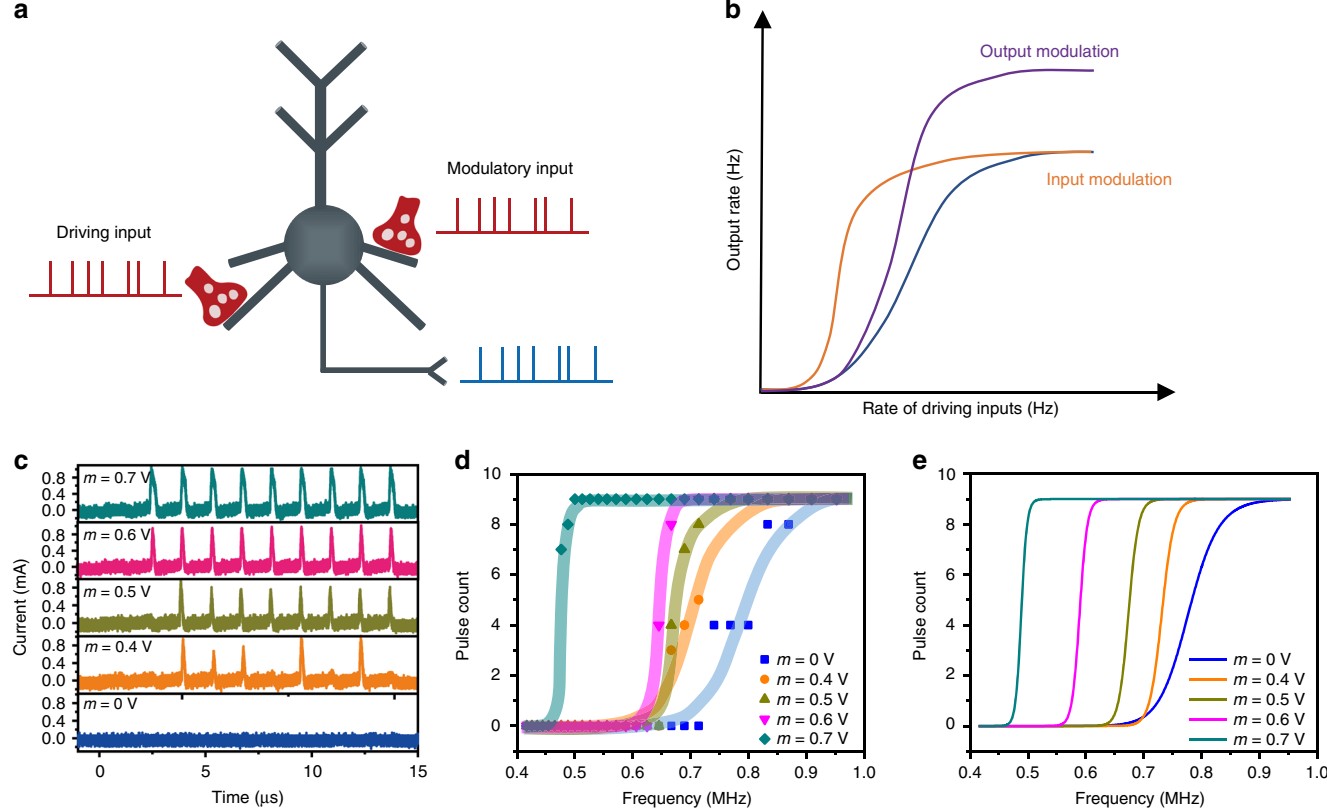

**Fig. 4 Neuronal gain modulation of NbO$_x$ based neuron. a** Schematic diagram of the rate-coded neuronal signaling. The driving input and modulatory input affect output firing rate jointly. **b** Schematic diagram of neuronal gain modulation by modulatory inputs, including input modulation (orange line) and output modulation (purple line). The blue line is the I–O curve with only the driving input. **c** Neuronal responses triggered by only the driving input (1.2 V amplitude, 1 μs width, 0.4 μs interval) applied on $S_1$ (bottom), and triggered by the driving input (1.2 V amplitude) and modulatory input (1 μs width, 0.4 μs interval) with different amplitudes of 0.4 V, 0.5 V, 0.6 V and 0.7 V, applied simultaneously to nodes $S_1$ and $S_2$, as shown in the middle and top of the panel. **d** Experimental neuron responses showing multiplicative gain modulation when applying different modulatory inputs ($m = 0, 0.4, 0.5, 0.6$ and 0.7 V). The background lines indicate the overall trend in data. **e** Fitting lines calculated by Eq. (2).

function of gain modulation. The gain modulation of a neuron is a roughly multiplicative or divisive change in the tuning curve of a neuron to a stimulus parameter, as other parameters or states change[16]. Such gain change is often observed in the response of cortical neurons and is thought to play an important role in neural computing[41]. There are also two forms of neuronal gain modulation through synaptic input. One is input modulation (orange line in Fig. 4b), where the rate-coded input–output (I–O) relationship is shifted along the x-axis upon altering the modulatory input, and the other is output modulation (purple line in Fig. 4b), where I–O relationship is shifted along the y-axis upon altering the modulatory input[17,36]. For input modulation, the maximum value of the I–O relationship does not change with modulation, but for output modulation, the maximum transmission rate increases or decreases proportionally (Fig. 4b).

To implement similar functions, a series of pulses (1.2 V in amplitude, 1 μs in width, 0.4 μs in interval, repeated for 10 cycles) were applied on $S_1$ as the driving input to the abovementioned NbO$_x$ neuron, and another pulse train (0, 0.4, 0.5, 0.6 or 0.7 V in amplitude, 1 μs in width, 0.4 μs in interval, repeated for 10 cycles) was applied on $S_2$ as the modulatory input, while the firing spike count is defined as neuronal output. Figure 4c shows the typical neural output in response to the driving and modulatory inputs with $m = 0, 0.4, 0.5, 0.6$ and 0.7 V, where $m$ is the amplitude of the modulatory input. One can see that the firing spike count of the neuron is 0 by only applying the driving input, while the firing spike count of the neuron is 5, 8, 9 and 9 by additional modulatory input with different amplitudes of 0.4, 0.5, 0.6 and

0.7 V. Figure 4d shows the experimental I–O relationships of the neuron as a function of driving input frequency when $m$ is adjusted, implying input gain modulation because I–O relationship is shifted horizontally. Furthermore, the I–O curves can be fitted by the following empirical functions written as[16,17]:

$$f(d, m) = R_{\max} \frac{(d \cdot a(m))^{n(m)}}{d_{50}^{n(m)} + (d \cdot a(m))^{n(m)}} \qquad (2)$$

$$a(m) = k_0 \cdot m^4 + k_1, \quad n(m) = k_2 \cdot m + k_3 \qquad (3)$$

where $d$ is the frequency of the driving input, $R_{\max}$ represents the maximum firing rate of the neuron, $f(d, m)$ represents the spike count of the neuron with both driving and modulatory inputs, $d_{50}$ is the frequency of the driving input that produces a half maximum response, $n$ is the Hill coefficient that determines the shape of the function and depends on the modulation input $m$. $k_0$, $k_1$, $k_2$, $k_3$, are fitting parameters. The gain of a neuron is thus the slope of the $f$-$d$ curve. The solid lines in Fig. 4e display the fitting results using Eqs. (2) and (3), where all the fitting parameters are fixed for the five $f$-$d$ curves, i.e., $k_0 = 1.11$, $k_1 = 0.45$, $k_2 = 100$, $k_3 = 50$, $R_{\max} = 9$, $d_{50} = 0.35$.

To further understand why Eq. (2) leads to a gain change, we define the gain as $\frac{\partial f}{\partial d}$. Taking the derivative of $d$ in Eq. (2), we

obtain an expression for the gain:

$$\frac{\partial f}{\partial d} = a(m) \cdot n(m) \cdot R_{\max} \cdot d_{50}^{n(m)} \cdot \frac{(d \cdot a(m))^{n(m)-1}}{(d_{50}^{n(m)} + (d \cdot a(m))^{n(m)})^2} \quad (4)$$

That is, the value of $\frac{\partial f}{\partial d}$ depends on $m$, and hence changes in the modulatory input can change its gain. More specifically, experimental results have demonstrated that the slope is increased upon application of a higher modulatory input (Fig. 4d), and hence the gain modulation has a multiplicative nature[17]. Such multiplicative modulation is widely used in visually guided reaching, collision avoidance, etc[42–44], and can now be realized in a single neuron, therefore enhancing its capability. This could pave the way for the construction of real intelligent systems.

**Pattern recognition in fully memristive neural networks**. Based on the abovementioned artificial neurons, we can replace the inputs with memristive synaptic devices, which can utilize the conductance changes of synaptic devices to enrich the functions, therefore forming a fully memristive neural network. Pt/Ta/Ta$_2$O$_5$/Pt synaptic devices (statistical electrical analysis of the synaptic device can be found in Supplementary Fig. 9 and Supplementary Note 4) and NbO$_x$ based neurons were directly fabricated and integrated together in hardware (Supplementary Fig. 10 and Supplementary Note 5), and Fig. 5a shows the overview of the structure consisting of synapse crossbar array with NbO$_x$ neurons in each row. The structural configurations of the NbO$_x$ neurons and Pt/Ta/Ta$_2$O$_5$/Pt synapses were assessed by SEM (Fig. 5b, d), cross-sectional high-resolution TEM (Fig. 5c, e) and corresponding energy-dispersive X-ray spectroscopy (EDS) characterization, including elemental mapping and line-scan as shown in Supplementary Figs. 11 and 12.

Such fully memristive neural network can be used for pattern recognition, as shown in Fig. 5f, g, where each neuron is connected to 4 input (pre-synaptic) neurons via synapses. Through offline training, the weights of the corresponding synapses can be programmed to desired values, and the fully memristor-based network, despite in small scale, can achieve pattern recognition from the neuronal output, as shown in Fig. 5f, g. Furthermore, supervised online learning was also conducted on the fully memristive neural network. In this case, a set of training data consisting of input patterns and expected output firing frequency of the neuron are shown to the network. Depending on the deviation between the expected and actual output, the synaptic weights are adjusted in an optimization process until the solution is best approximated and the network is trained similarly. We utilized the network shown in Fig. 5h with a training method shown in Supplementary Fig. 13a and b to illustrate an example of supervised learning based on simplified δ-rule (see Supplementary Note 6). Figure 5i shows the evolutions of the four synaptic weights over time during supervised learning by a single neuron. Initially, the synapses $S_2$ and $S_3$ were in high conductance state, while the $S_1$ and $S_4$ were in low conductance state, namely, "0110". When the input pattern "1010" was repeated, by constantly adjusting the weight of the synapses, the neuron adapted to it over time, until finally the neuron had learned this pattern. As shown in Supplementary Fig. 13c, it can be clearly seen that the neuron has been successfully trained, since it recognizes "1010", indicating that our neuromorphic system can solve simple image recognition tasks.

**Large-scale fully memristive spike-encoded neural networks**. To show the potential of the present devices in the construction of large-scale fully memristive spiking neural networks (SNN), we performed a simulation based on experimental data. Figure 6a illustrates a 3-layer spiking neural network, which is composed of

784 input neurons, 100 hidden neurons and 10 output neurons, where the 784 inputs and 10 outputs correspond to a MNIST data size of $28 \times 28$ and 10 possible classes (from 0 to 9), respectively. We evaluate the network performance by MNIST handwritten digit classification, and detailed simulation process is shown in Fig. 6b, where the memristive SNN is trained online by backpropagation based on experimentally measured electrical characterics of TaO$_x$ synapses (Supplementary Fig. 14a, Supplementary Note 7) and the NbO$_x$ neuron (Supplementary Note 7). As the first step in forward pass, a $28 \times 28$ MNIST image is converted to a $1 \times 784$ vector, and each pixel value is used to generate a Poisson event, i.e., a random voltage spike. The spiking possibility is higher if the corresponding pixel value is larger. During the simulation lasting for 500 time steps, there is a 784-spike-event vector at every time step and a train of 500 spikes for each input neuron. These spike trains are then fed into the memristor crossbar and converted to weighted current sums through the columns. A row of transimpedance amplifier can be used to amplify and convert the currents to analog voltages of (−2, 2 V). The neurons can then integrate the analog voltages and generate spikes when reaching the firing threshold, which propagate to the next layer for similar processes. At last, the spiking numbers of output neurons are counted, and the index of most frequently spiking neuron is taken as the prediction result. During the backward propagation of errors, since the input–output function of a spiking neuron is a step function, which has infinite gradient, it is replaced by a soft activation function, e.g., sigmoid in the present case, to get the gradient and then the synaptic weights are adjusted accordingly (see Supplementary Note 7 for more details). Such training is performed for 100 epochs (Supplementary Fig. 14c).

Figure 6c shows that the 10 output neurons learnt specific digits during the training process, and depicts also the statistics of the firing numbers issued by 10 neurons in the case of input pictures corresponding to the category numbers themselves, where the input picture is correctly identified in most cases. The 3-layer SNN reaches a training accuracy of 81.45% after 100 epochs (Supplementary Fig. 14c), and the classification accuracy of the simulated network is 83.24% on MNIST test dataset. The inference latency can be as low as 10 time steps, which is 10 μs, with each neuron representing prediction result firing ~3.5 spikes on average. Figure 6d further shows a confusion matrix of the testing results from the 10,000 MNIST test dataset. As a measure on the classification accuracy, the confusion matrix in Fig. 6d displays the classification result in each column while the expected (actual) result in each row, where the number of instances is depicted by the color bar. As a result, the confusion matrix allows direct visualization of the firing number distribution of the trained output neurons in response to the test inputs, demonstrating that the test inputs are well classified after training.

**Coincidence detection in fully memristive neural networks**. Besides pattern recognition/classification, the rich nonlinear dynamics of the artificial neuron (Figs. 3 and 4) in the present work can give rise to more complex functions at the network level. Since neurons transmit information by spike sequences, which carry essential dynamic characteristics of neurons and features of the stimuli, coincidence detection has been found to be a highly efficient information processing function and has great significance in both auditory[45,46] and visual systems[47], where synchronous synaptic inputs are preferably transmitted. Such coincidence detection can be achieved based upon the spatio-temporal dynamics of the NbO$_x$ neuron, as shown in Fig. 7. A $2 \times 1$ array consisting of two Pt/Ta/Ta$_2$O$_5$/Pt synapses and a Pt/Ti/NbO$_x$/Pt/Ti neuron was employed to detect whether the two

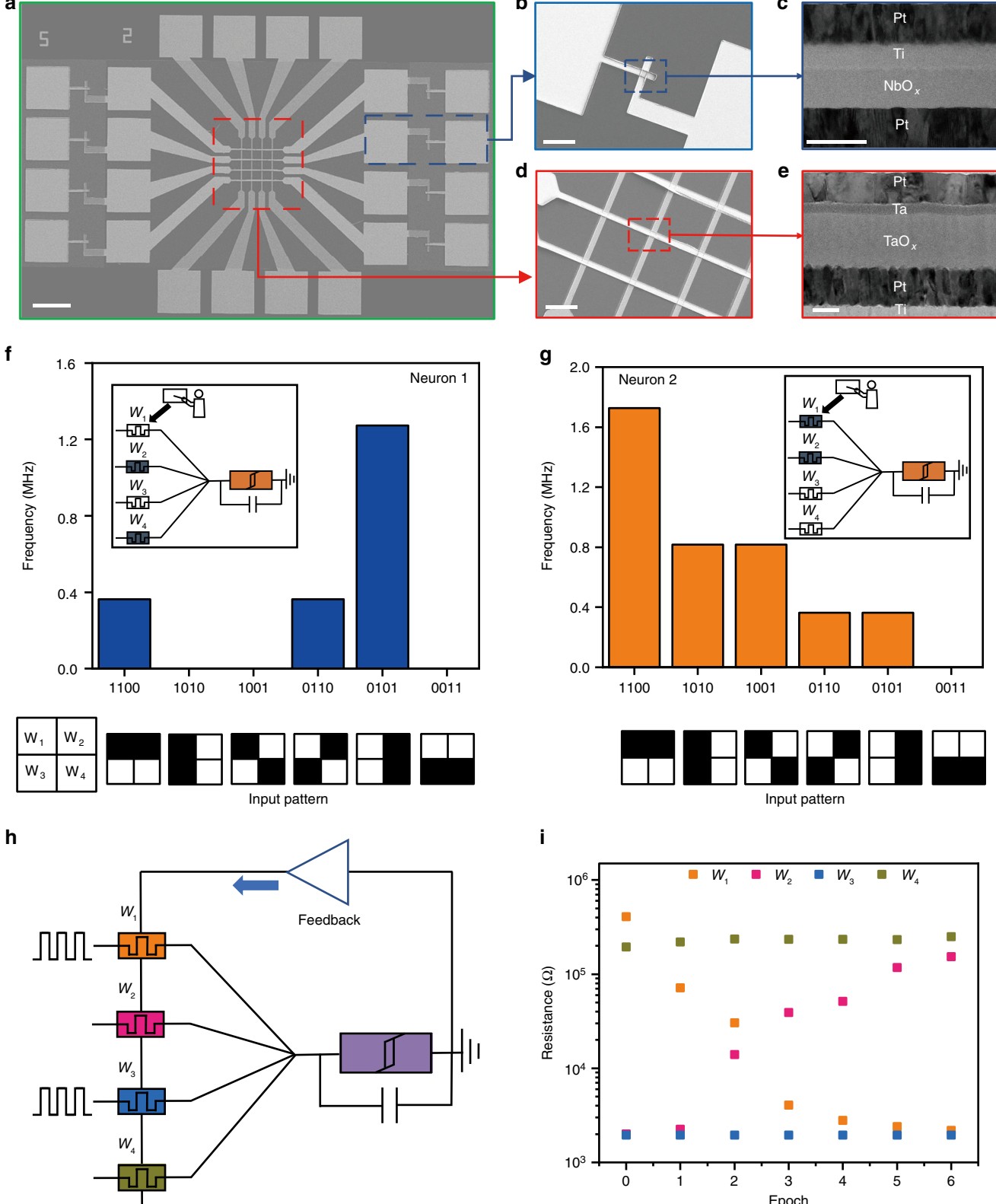

**Fig. 5 Fully memristive spiking neural network. a** Scanning electron microscopy (SEM) image of the monolithically integrated memristive neural network. Scale bar, 100 μm. **b** SEM image of the Pt/Ti/NbO$_x$/Pt threshold switching device. Scale bar, 20 μm. **c** Cross-sectional transmission electron microscopy (TEM) image of the Pt/Ti/NbO$_x$/Pt threshold switching device. Scale bar, 50 nm. **d** SEM image of a Pt/Ta/TaO$_x$/Pt synapse device. Scale bar, 10 μm. **e** Cross-sectional TEM image of the Pt/Ta/TaO$_x$/Pt device. Scale bar, 20 nm. **f, g** Neuronal outputs when presented with different input patterns. **f** Neuron 1 learnt to recognize pattern "0101", while neuron 2 learnt to recognize pattern "1100" **g**. The input pulses are 1 V in amplitude, 1 μs in width, 0.1 μs in interval and repeated for 10 cycles. **h** Schematic illustration of supervised learning in a fully memristive neural network. **i** Evolutions of synaptic weights over time during online learning, where the pattern "1010" is repeatedly applied, with an initial state of "0110".

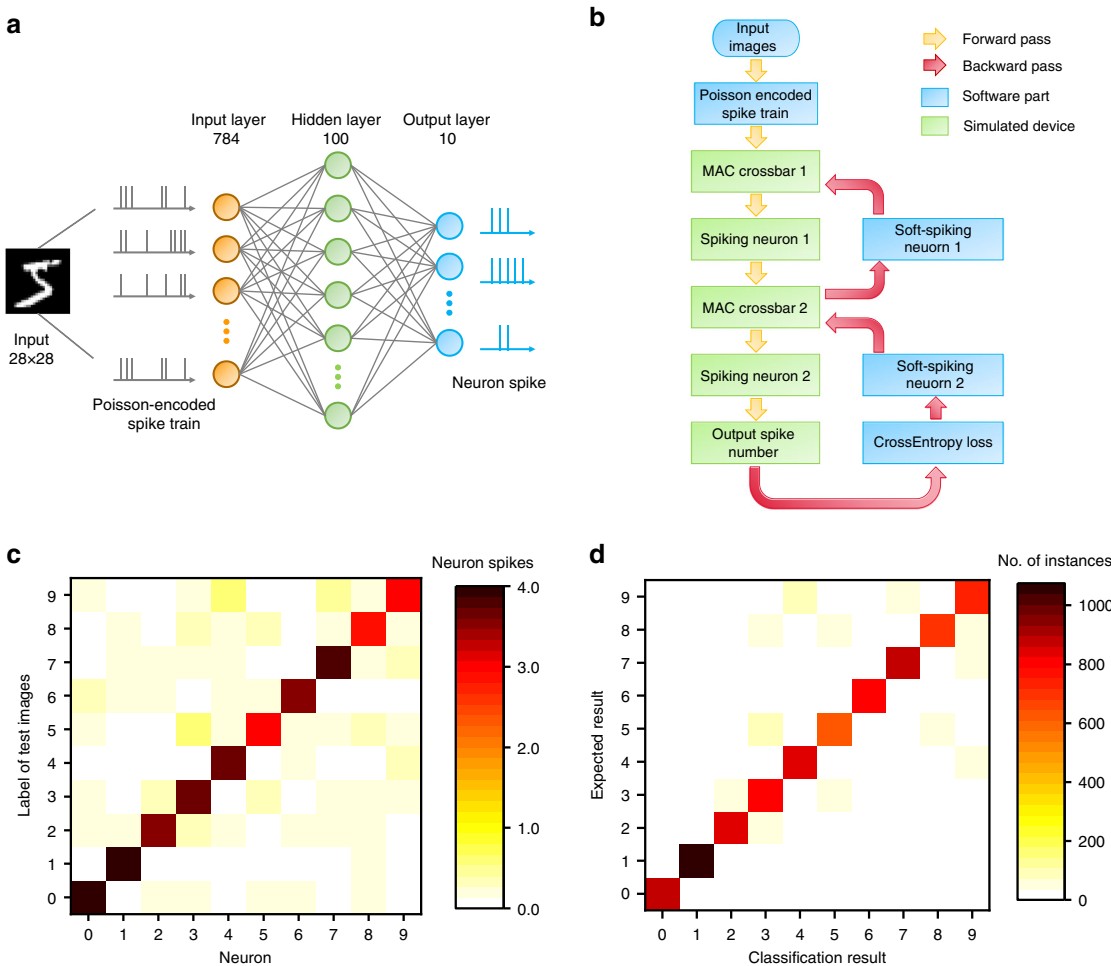

**Fig. 6 A 3-layer spiking neural network by simulation. a** Schematic of the spiking neural network for MNIST classification. Input images are first converted to Poisson spike trains, where the spiking rates are proportional to pixel values. The spike trains are then input to the network, weighted and integrated by the neurons. At last, we count the spiking rates of output neurons to get the prediction results. **b** Flow chart of the simulation process. In the forward phase (yellow arrow), input spike trains are weighted by the memristor crossbar and then integrated on neurons, and the spiking rates of output neurons are used in loss computing. In the backward phase (purple arrow), a soft function like sigmoid is used as an alternative to neuron spike function (step function) in gradient computing. The computing units in green boxes are simulated based on experimental data, while the units in blue boxes are implemented by software. **c** In the case where the input picture is correctly identified, the statistics of the firing numbers issued by ten neurons in the case of input pictures corresponding to the MNIST numbers themselves. Statistics of firing numbers of the category neurons in response to different inputs, showing that the inputs are classified correctly in most cases. **d** Averaged confusion matrix of the testing results, showing that the test inputs are well classified after training.

presynaptic inputs were synchronized or not (Fig. 7a). As shown in Fig. 7b, when the two presynaptic inputs (1.1 V in amplitude, 1 μs in width, 1.5 μs in interval, for 50 cycles) were simultaneously applied into the network, the output neuron was activated with a firing rate of ~0.3 MHz. When the two presynaptic inputs were asynchronous ($S_2$ behind $S_1$ by 1.2 μs) or randomly arranged in time (where $S_1$ is a periodic input and the timing of $S_2$ is random relative to $S_1$), the neuron cannot fire (Fig. 7c, d). These results are consistent with the spatiotemporal dynamics of the neuron in Fig. 3k, l and imply the potential of the fully memristive neural network in coincidence detection.

To further extend the applicability of the neuronal spatiotemporal dynamics in large-scale neural computation, we have simulated a spiking neural network using Brian2 simulator[48], where the neuronal parameters were extracted from electrical measurements (Supplementary Fig. 15, Supplementary Note 8) and the neuron receives 4000 excitatory and 1000 inhibitory spike trains following Poisson statistics, as shown in Fig. 7e. The excitatory and inhibitory spikes are 0.136 V and −0.134 V in

amplitude, respectively, and the average rate of Poisson input is 10 kHz. Previous studies have shown that when excitation and inhibition are balanced, synchrony in a very small proportion of random inputs can lead to dramatic increase in the firing rate of the output neuron[49]. In the present case synchronous events are included as an independent Poisson process at 400 kHz, and only 15 excitatory spike trains (0.3% in proportion) are randomly picked to simultaneously fire for each event (Fig. 7f). Simulation results show that the firing rate of the neuron can be increased by over 10 times as a result of the synchronous events, as shown in Fig. 7g, h, therefore indicating the potential of the present neuronal dynamics in detecting fine temporal correlations in massive signals and small timescales.

**Gain modulation enabling receptive field remapping**. The gain modulation of the spiking neuron (Fig. 4) in the present work can also enrich the functionality of memristive neural networks. In biology, gain modulation is a canonical neural computation

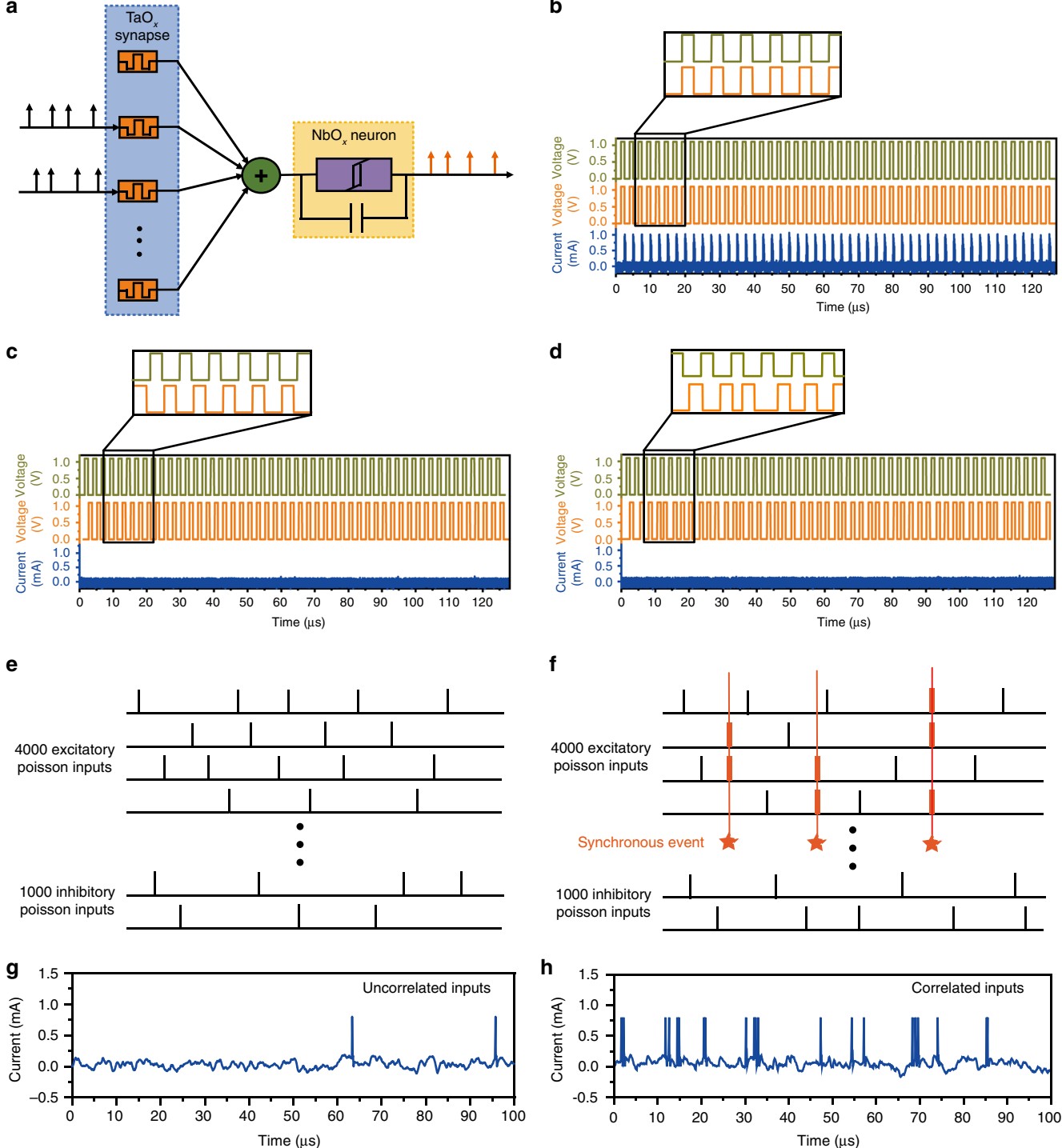

**Fig. 7 Coincidence detection based on spatiotemporal dynamics. a** Schematic diagram of coincidence detection in the fully memristive neural network. **b** The neuronal response by two synchronous input pulse trains (1.1 V in amplitude, 1 μs in width, 1.5 μs in interval, for 50 cycles) applied on $S_1$ and $S_2$. **c** The neuronal response by two asynchronous input pulse trains (1.1 V in amplitude, 1 μs in width, 1.5 μs in interval, for 50 cycles) applied on $S_1$ and $S_2$, where $S_2$ is behind $S_1$ by 1.2 μs. **d** The neuronal response by two asynchronous input pulse trains (1.1 V in amplitude, 1 μs in width, for 50 cycles) applied on $S_1$ and $S_2$, where $S_1$ has and interval of 1.5 μs and $S_2$ is random relative to $S_2$ in timing. **e** Input pulse trains of the neuron from independent 4000 excitatory and 1000 inhibitory random spike trains following Poisson statistics. **f** Introduction of synchronous events following Poisson statistics into the excitatory inputs, where the input rates are unchanged and the proportion of synchronous events is 0.3%. **g** Simulated neuronal response as a result of the inputs shown in **e**, where the artificial neuron only fires 2 spikes. **h** Simulated neuronal response as a result of the inputs shown in **f**, where the artificial neuron fires 21 spikes.

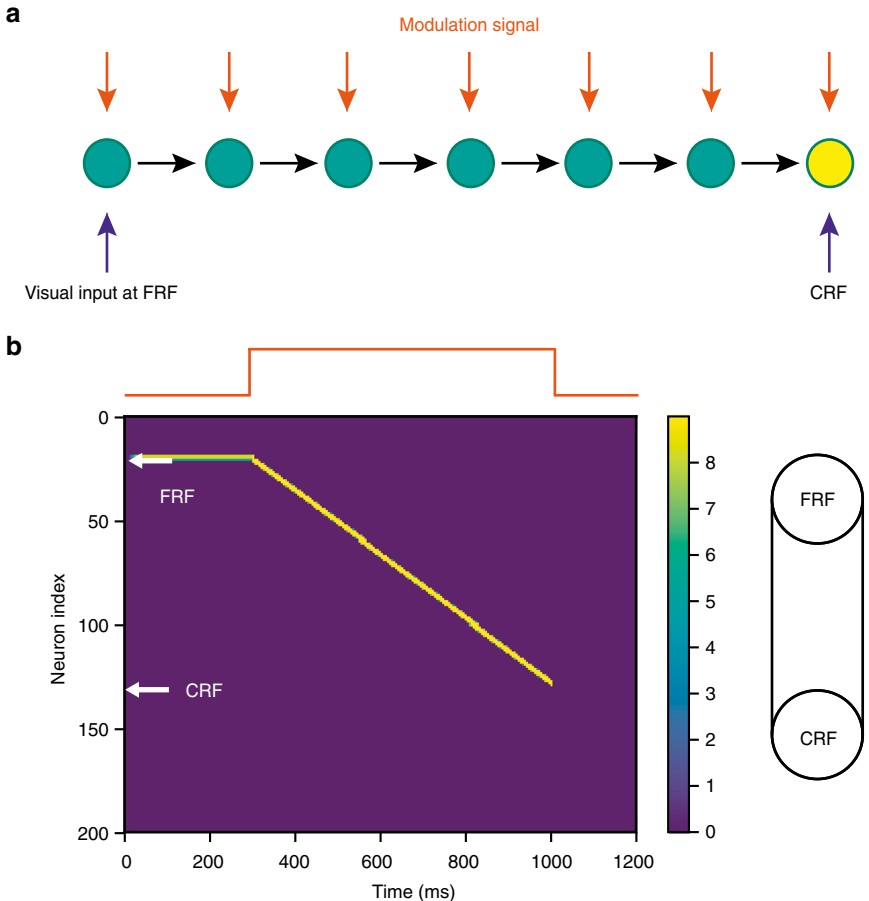

**Fig. 8 A network of memristive neurons with gain modulation for receptive field remapping. a** A one-dimensional network where neurons are connected in an uni-direction. Each neuron receives a modulatory signal (orange arrow). A visual input is applied at the future receptive field (FRF) of a neuron (the yellow one). **b** The visual input at FRF and the modulatory signal (top, red line) triggers a propagation of neural activity from FRF to current receptive field of the yellow neuron, as if the receptive field of the yellow neuron is temporally expanded when the modulatory signal is applied, achieving RF remapping (right panel). The distance of remapping is controlled by the duration of the modulatory signal. In the simulation, the visual input is applied during 0−300 ms, while the modulatory signal $m = 0.8$ is applied during 300−1000 ms. $\theta_{\mathrm{thr}} = 0.41$, $W_{i,i-1} = 0.095$, $\tau = 10$ ms, $dt = 1$ ms, $I_{\mathrm{ext},0} = 0.09$.

involved in many brain functions[17] and plays a key role in maintaining visual stability when human eyes make frequent saccadic movements[50]. Such stability is achieved by temporal expansion of receptive fields (RFs) of the neurons encoding object locations, named RF remapping, to compensate the shift of retinal locations of objects caused by saccade, such that the neural system will not perceive abrupt visual changes during the saccade. The above RF remapping is enabled by gain modulation, where an efference copy of the saccade serves as a gain signal that in turn triggers temporal propagation of the neuronal responses between current RF (CRF) and future RF (FRF)[50]. The abovementioned receptive field remapping could significantly enhance the stability of artificial visual systems if it can be implemented in hardware, since similar instability issue can arise in real world, for example image shifts due to camera shaking may be confounded with real movements of objects.

Based upon the memristive neurons with gain modulation (Fig. 4), a one-dimensional network model is constructed, which includes 200 neurons connected in an uni-direction, as shown in Fig. 8a. Denote $d_i$ as the input to neuron $i$, whose dynamics is given by:

$$\tau \frac{dd_i}{dt} = -d_i + W_{i,i-1}[f(d_{i-1}, m(t)) - \theta_{\mathrm{thr}}]_+ + I_{\mathrm{ext},i} \qquad (5)$$

where $\tau$ is the time constant, $W_{i,\ i-1}$ is the connection strength from neuron $i-1$ to $i$, $I_{\mathrm{ext},\ i}$ is the visual input, and $\theta_{\mathrm{thr}}$ is a threshold. The symbol $[x]_+ = x$ for $x > 0$, and otherwise $[x]_+ = 0$. Only above $\theta_{\mathrm{thr}}$, i.e. $f(d_i, m) > \theta_{\mathrm{thr}}$, the neuronal response can propagate to the neighborhood. By this, the modulation signal $m$ controls the propagation of network activity. The firing rate of the neuron is given by Eq. (2), and the input modulation takes the multiplicative form as shown in Fig. 4.

The simulated network clearly shows that a visual input at FRF can only evoke a weak response of the neuron without modulatory signals (Fig. 8d). In stark contrast, when a modulatory signal is applied, it triggers a propagation of neuronal responses from FRF to CRF, as if the receptive field of the neuron is temporally expanded (Fig. 8b), which disappears when the modulatory signal terminates. Such result therefore demonstrates that memristive networks consisting of the spiking neurons with gain modulation can achieve RF remapping for enhanced visual stability, implying the great potential of memristive neural networks in neuromorphic computing.

## Discussion

We have demonstrated an artificial neuron based on threshold switching in NbO$_x$ devices, which displays four critical features: threshold-driven spiking, spatiotemporal integration, dynamic logic including XOR that is not linearly separable and gain modulation. Such spiking neuron is more capable compared with existing functions described by a simple point neuron model.

Compared with existing approaches for building artificial neurons, the NbOx based neurons exhibit fast speed and comparable power consumption (see detailed comparison in Supplementary Table 2, Supplementary Note 9). Furthermore, latest mechanistic insights into NbOx have revealed that the threshold switching might be achieved with lower switching temperature and hence reduced power consumption[30,31], which implies significant room for further device optimization based on NbOx. Such artificial neuron with spatiotemporal dynamics and gain modulation could form an integral part for biologically plausible, fully memristive neural networks, where both the neurons and synapses are built by compact memristive devices, therefore holding great prospect for intelligent neural networks that alleviate the currently high requirements for chip area and electric power. A 4 × 4 fully memristive spiking neural network consisting of volatile NbOx memristor-based neurons and nonvolatile TaOx memristor-based synapses in the same crossbar array is experimentally demonstrated, showing capability in performing pattern recognition through online learning based on a simplified δ-rule and coincidence detection through experiments and simulations. Moreover, the multiplicative gain modulation of the artificial neuron forms an enabling factor in receptive field remapping based on memristive neural networks, which can significantly enhance the stability of artificial visual systems. A 3-layer fully memristive spiking neural network is also constructed by simulation, which achieves a test accuracy of 83.24% in MNIST handwritten digit classification. This study could offer an alternative energy-efficient, bio-faithful route toward hardware implementation of neuromorphic computing systems.

## Methods

**Fabrication of NbOx threshold switching devices.** All NbOx devices studied in this work were fabricated on SiO₂/Si substrates. First, bottom electrodes consisting of 30 nm thick Pt film with 5 nm thick Ti adhesion layer were deposited by e-beam evaporation and patterned by photo lithography and lift-off processes. Next, we deposited NbOx film (30 nm) serving as the switching layer on Pt film by DC reactive sputtering using Ar and O₂ mixture gases (Ar: 10 mTorr, O₂: 70 mTorr). Finally, 10 nm thick Ti was deposited as the top electrode and capped by 30 nm thick Pt protection layer, where the patterning of the top electrodes was done by photo lithography and lift-off processes.

**Fabrication of the fully memristive neural network.** The fully memristive neural network was fabricated on SiO₂/Si substrates. First, 30 nm thick Pt film with 5 nm thick Ti adhesion layer of 2 μm width was deposited by e-beam evaporation, where the patterning of the bottom electrodes was done by photo lithography and lift-off processes. Afterward, photo lithography was used to form the patterns of the switching layer, followed by RF sputtering and lift-off processes. The switching layer of synaptic devices in this study was Ta₂O₅ (30 nm). Subsequently, a Ta (10 nm)/Pt (30 nm) of 2 μm width as a middle electrode (ME) was formed to cover the BE vertically, where the patterning of the middle electrodes was done by photo lithography and lift-off processes, which resulted in Pt/Ta/Ta₂O₅/Pt/Ti synaptic devices. Next, photo lithography was performed to define the switching layer of the artificial neurons, and NbOx film (30 nm) was deposited as the switching layer by DC reactive sputtering using Ar and O₂ mixture gases (Ar: 10 mTorr, O₂: 70 mTorr). Finally, 10 nm thick Ti was deposited as the top electrode and capped by 30 nm thick Pt protection layer by DC sputtering, where the patterning of the top electrodes was done by photo lithography and lift-off processes. Detailed flowchart of the preparation process is given in Supplementary Fig. 10.

**Electrical measurements.** Electrical measurements were performed using an Agilent B1500A semiconductor parameter analyzer and the Agilent DSO90254A digital storage oscilloscope. Voltage pulses were applied by the Agilent B1500A. One channel of the oscilloscope was used to measure the voltage at the output of the Agilent B1500A, while another channel was used to measure the current of total circuit. In Fig. 2f, we used a 3.6 kΩ resistor and a 30 pF capacitor with a 0.1 μs pulse interval, 1 μs pulse width, and 20 pulse cycles to study the dependence of pulse count on pulse amplitude; to study the dependence of pulse count on pulse width, we used a 3.6 kΩ resistor and a 30 pF capacitor with 0.1 μs pulse interval, 1.3 V pulse amplitude, and 20 pulse cycles; to study the dependence of pulse count on the pulse interval, we used a 3.6 kΩ resistor and a 30 pF capacitor with 1.3 V pulse amplitude, 1 μs pulse width, and 20 pulse cycles; to study the dependence of pulse count on $R_L$, we used a 30 pF capacitor with 1.3 V pulse amplitude, 1 μs pulse width, 0.1 μs pulse interval, and 20 pulse cycles; to study the dependence of pulse

count on $C_m$, we used a 3.6 kΩ resistor with 1.3 V pulse amplitude, 1 μs pulse width, 0.1 μs pulse interval, and 20 pulse cycles. The other experimental circuit parameters are shown in Supplementary Information.

**Microstructural and compositional characterization.** The TEM samples in this work were prepared by the focused ion beam (FIB) technique using a dual-beam FIB system (FEI Helios Nanolab workstation). During FIB patterning, the sample was first coated by a Pt layer deposited using the electron beam to avoid surface damages, followed by higher-rate Pt coating using normal ion beam process that served as majority of the protective layer during FIB cutting. TEM and STEM images as well as EDS measurements were performed on FEI Tecnai F20. The SEM characterization was conducted on a field emission SEM (Merlin Compact).

**Simulations.** The three-layer SNN with an architecture of 784 × 100 × 10 was simulated in Python, where the weight matrices and neurons were implemented by the TaOx memristor crossbar and NbOx devices, respectively. Large-scale simulation on coincidence detection was based on Python3 and Brain2, where the parameters of artificial neurons were extracted from experimental data by curve fitting in MATLAB (Supplementary Fig. 15). During the simulation process, each neuron received independent 4000 excitatory and 1000 inhibitory random spike trains following Poisson statistics at 10 kHz, and synchronous events were included as an independent Poisson process at 400 kHz. The simulation on receptive field remapping with gain modulation was based on Python. The network was modeled as a dynamic system, where a first-order Euler approximation with time step $\Delta t$ is used:

$$x_i(t+1) = (1-\alpha)x_i(t) + \alpha(W_{i,i-1}[f(x_{i-1}(t), m(t)) - \theta_{thr}]_+ + I_{ext,i}(t))$$

where $\alpha = \frac{\Delta t}{\tau}$, $\Delta t = 1$ ms.

## Data availability

All data supporting this study and its findings are available within the article, its Supplementary Information and associated files. The source data underlying Figs. 2b, c, e, f, 3c–e, g–i, k, l, 4c, d, 5f, g, i, 6c, d, 7b–d, g, h and Supplementary Figs. 2–8, 9b–d, 11b, 12b, 13c, 14, 15 have been deposited at [https://doi.org/10.5281/zenodo.3875763].

## Code availability

The codes used for the simulations are described in [https://github.com/duanqingxi/NCOMMS-20-02976.git] or are available from the corresponding author upon reasonable request.

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

## Acknowledgements

The authors thank Liutao Yu for helpful discussions. This work was supported by the National Key R&D Program of China (2017YFA0207600), National Outstanding Youth Science Fund Project of National Natural Science Foundation of China (61925401), Project 2019BD002 supported by PKU-Baidu Fund, National Natural Science Foundation of China (61674006, 61927901, and 61421005), and the 111 Project (B18001). Y.Y. acknowledges the support from the Fok Ying-Tong Education Foundation, Beijing Academy of Artificial Intelligence (BAAI) and the Tencent Foundation through the XPLORER PRIZE.

## Author contributions

Q.D. and Y.Y. designed the experiments. Q.D. fabricated the devices. Q.D., K.Y. and T.Z. performed electrical measurements. Z.J., X.Z., Y.W. and S.W. performed the simulations. Q.D., Z.J., X.Z., Y.W. and Y.Y. prepared the manuscript. Y.Y. and R.H. directed all the research. All authors analyzed the results and implications and commented on the manuscript at all stages.

## Competing interests

The authors declare no competing interests.
