## [Peer Review File · Nature Communications]

Manuscript ID: NCOMMS-20-02976

Title: Spiking Neurons with Spatiotemporal Dynamics and Gain Modulation for Monolithically Integrated Memristive Neural Networks

Comments from Reviewer #1

Overall Remarks: **Development of fully integrated neuromorphic systems is a topic of very wide interest and the presented work has taken important steps in that direction. Authors have demonstrated a NbO_x based memristor as artificial neuron which can handle spatiotemporal input dynamics and can exhibit output gain modulations. They have also demonstrated integrated neuron and synapse in a single crossbar array with capability of pattern recognition through supervised learning.**

The presented data and interpretation is detailed and convincing. I recommend the work for publication in Nature Communications.

Our response: We would like to sincerely thank the reviewer for his/her positive comments, which summarize the key findings of our work. We are particularly pleased to see the reviewer's comment in "*Development of fully integrated neuromorphic systems is a topic of very wide interest*" and "*the presented work has taken important steps in that direction*". The reviewer has recommended this work for publication.

Comments from Reviewer #2

Overall Remarks: **The manuscript "Spiking Neurons with spatiotemporal dynamics and gain modulation for monolithically integrated memristive neural networks" reports that the NbO_x based volatile memristor can be used to construct an oscillatory neuron for neural networks. The major contribution of this work is the experimental demonstration of spatiotemporal dynamics and the gain modulation of the spiking neurons. These two features are highly desired for spiking neuromorphic applications, and therefore the topic is of high interest of**

both neuroscience and electrical engineering community.

Our response: We would like to sincerely thank the reviewer for pointing out the major contribution of this work and significance of the subject. We have carefully considered the reviewer's advices, performed new experiments/simulations (revised Figs. 3, 4 & 6, new Figs. 7 & 8, new Supplementary Figs. 2, 3, 5, 6, 7, 8, 14c & 15) and thoroughly revised the manuscript to address all the questions, as can be found below.

1. The major concern from the reviewer is that the authors did not seem to provide enough linkage between their fully memristive neural networks (both experiment and simulation) and the spatiotemporal dynamics and gain modulations of the artificial neuron. The inference demonstration for the pattern recognition with offline trained weight, in the view of the reviewer, is a simple linear model (e.g. single layer perception) and therefore is only connected with the spatial summation of the proposed neuron. The similar ideas were proposed and demonstrated previously in cited reference 23, 24. The reviewer believes the manuscript could be a very interesting and exciting one if the authors demonstrate fully memristive neural networks that make use of the rich nonlinear dynamic responses of the artificial neuron, to be consistent with the major claims of this manuscript.

Our response: We would like to thank the reviewer for raising this important question. In order to bridge the rich nonlinear dynamics of the artificial neurons in the present work with functions of fully memristive neural networks, we have further utilized the spatiotemporal dynamics for coincidence detection in fully memristive neural networks. A 2×1 array consisting of two Pt/Ta/Ta₂O₅/Pt synapses and a Pt/Ti/NbO_x/Pt/Ti neuron was employed to detect whether the two pre-synaptic inputs are synchronized or not, and the experimental results are displayed in a new figure Fig. 7a–d. One can see that when the pre-synaptic inputs are synchronized (Fig. 7b), asynchronous (Fig. 7c) or randomly arranged in time (Fig. 7d), the NbO_x neuron exhibited different firing rate, which is based on the spatiotemporal dynamics of the neuron shown in Fig. 3k,l. We

have further simulated a large-scale spiking neural network, where the parameters of the neuron were extracted from electrical experiments (Supplementary Fig. 15, Supplementary Note 8) and the neuron receives 4000 excitatory and 1000 inhibitory spike trains following Poisson statistics. Once 15 out of the 4000 excitatory inputs (<1%) were randomly selected to include synchronous events, the firing rate of the neuron can be increased by over 10 times, once again implying the applicability of the present neuronal dynamics in detecting fine correlations in massive signals and small timescales.

We have included these new results on coincidence detection in fully memristive neural networks in the new Fig. 7 (appended below for the reviewer's convenience), along with further discussion in Page 24–25. In addition, the gain modulation characteristics of the artificial neuron have also been utilized to achieve receptive field remapping, which can potentially enhance the stability of artificial visual systems, as shown in the response to question #2 of the reviewer.

Page 24–25: *“Besides pattern recognition/classification, the rich nonlinear dynamics of the artificial neuron (Figs. 3 and 4) in the present work can give rise to more complex functions at the network level. Since neurons transmit information by spike sequences, which carry essential dynamic characteristics of neurons and features of the stimuli, coincidence detection has been found to be a highly efficient information processing function and has great significance in both auditory^{45,46} and visual systems⁴⁷, where synchronous synaptic inputs are preferably transmitted. Such coincidence detection can be achieved based upon the spatiotemporal dynamics of the NbO_x neuron, as shown in Fig. 7. A 2×1 array consisting of two Pt/Ta/Ta₂O₅/Pt synapses and a Pt/Ti/NbO_x/Pt/Ti neuron was employed to detect whether the two pre-synaptic inputs were synchronized or not (Fig. 7a). As shown in Figure 7b, when the two pre-synaptic inputs (1.1 V in amplitude, 1 μs in width, 1.5 μs in interval, for 50 cycles) were simultaneously applied into the network, the output neuron was activated with a firing rate of ~0.3 MHz. When the two pre-synaptic inputs were asynchronous (S₂ behind S₁ by 1.2 μs) or randomly arranged in time (where S₁ is a periodic input and the timing of*

S_2 is random relative to S_1), the neuron cannot fire (Fig. 7c,d). These results are consistent with the spatiotemporal dynamics of the neuron in Fig. 3k,l and imply the potential of the fully memristive neural network in coincidence detection.

To further extend the applicability of the neuronal spatiotemporal dynamics in large-scale neural computation, we have simulated a spiking neural network using Brian2 simulator⁴⁸, where the neuronal parameters were extracted from electrical measurements (Supplementary Fig. 15, Supplementary Note 8) and the neuron receives 4000 excitatory and 1000 inhibitory spike trains following Poisson statistics, as shown in Fig. 7e. The excitatory and inhibitory spikes are 0.136 V and -0.134 V in amplitude, respectively, and the average rate of Poisson input is 10 kHz. Previous studies have shown that when excitation and inhibition are balanced, synchrony in a very small proportion of random inputs can lead to dramatic increase in the firing rate of the output neuron⁴⁹. In the present case synchronous events are included as an independent Poisson process at 400 kHz, and only 15 excitatory spike trains (<1% in proportion) are randomly picked to simultaneously fire for each event (Fig. 7f). Simulations results show that the firing rate of the neuron can be increased by over 10 times as a result of the synchronous events, as shown in Fig. 7g,h, therefore indicating the potential of the present neuronal dynamics in detecting fine temporal correlations in massive signals and small timescales.”

Figure 7. Coincidence detection based on spatiotemporal dynamics. (a) Schematic diagram of coincidence detection in the fully memristive neural network. (b) The neuronal response by two synchronous input pulse trains (1.1 V in amplitude, 1 μs in width, 1.5 μs in interval, for 50 cycles) applied on S_1 and S_2 . (c) The neuronal response by two asynchronous input pulse trains (1.1 V in amplitude, 1 μs in width, 1.5 μs in interval, for 50 cycles) applied on S_1 and S_2 , where S_2 is behind S_1 by 1.2 μs . (d) The neuronal response by two asynchronous input pulse trains (1.1 V in amplitude, 1 μs in width, for 50 cycles) applied on S_1 and S_2 , where S_1 has an interval of 1.5 μs and S_2 is random relative to S_1 in timing. (e) Input pulse trains of the neuron from independent 4000 excitatory and 1000 inhibitory random spike trains following Poisson statistics. (f) Introduction of synchronous events following Poisson statistics

into the excitatory inputs, where the input rates are unchanged and the proportion of synchronous events is 0.3%. (g) Simulated neuronal response as a result of the inputs shown in (e), where the artificial neuron only fires 2 spikes. (h) Simulated neuronal response as a result of the inputs shown in (f), where the artificial neuron fires 21 spikes.

2. The other major concern is that the data shown in Fig. 4d does not seem to be enough to support the gain modulation claim. The data points shown in Fig. 4d does not seem to be conclusive to support the empirical equation (Eq. 2, 3) since there are only two data points when the neuron fires without modulation input (blue curve). It is also not very clear to the reviewer how the driving input and modulatory input connect to the artificial neuron; therefore, the reviewer suggests using a circuit diagram schematic here to avoid confusion. The reviewer's understanding is that the physical connection is shown in supplementary figure 3, where R_1 (or R_2) is connected to the driving input and R_2 (or R_1) to modulatory input. In this case, the intuition would be that the two load resistors (or synaptic memristors) that are connected in parallel performing an additive operation to the input, which is also the foundation of the following neural networks demonstrations by the authors. This intuition can also be supported by the data provided in Fig. 4d, where the modulatory input of 0.6 V and 0.7 V does not seem to change the shape of the curve but changed the minimum synaptic input that generates neuron fire event (rheobase), which contradicts the multiplicative claim (see also the reference by Silver et al). The reviewer strongly suggests the authors provide more persuasive evidence to support the claim that the modulatory input is multiplicative and make use of this feature to build a functional neural network.

Our response: We would like to thank the reviewer to raising this important question. In light of his/her advice, we have included a circuit diagram in revised Fig. 3a, where R_1 and R_2 are connected to the driving input and modulatory input, respectively (or vice versa). Detailed experimental parameters for R_1 , R_2 and C_m in all the measurements shown in Figs. 3 and 4 are described in detail in Supplementary Table 1.

In Fig. 4d, although the shapes of the curves corresponding to modulatory inputs of 0.6 V and 0.7 V appear to be similar due to the similar voltage amplitudes, their shapes especially the slopes are dramatically different from the case without modulatory input ($m = 0$), thus demonstrating a change in neuronal gain. Since the slopes are increased upon application of higher modulatory inputs, the gain modulation has a multiplicative nature (Ref. 17). But we appreciate the reviewer’s valuable advice and fully agree that the data quality originally in Fig. 4d are not perfect, so we have performed additional experiments and revised Fig. 4, in order to examine the nature of the neuronal gain more clearly. The further experimental results unambiguously show that the slope of f - d curve increases as the amplitude of modulatory inputs increases (Fig. 4d), demonstrating that the neuronal gain modulation has a multiplicative nature (Ref. 17). In addition, we have revised the model in Eqs. (2–4) slightly to better fit the new experimental data (Fig. 4e), which has the same input gain modulation property as described in the previous model.

The multiplicative gain modulation of the artificial neuron can be an enabling factor in receptive field remapping based on memristive neural networks, which can significantly enhance the stability of artificial visual systems. We have constructed a neural network in simulation based on the experimentally verified gain modulation behavior in Fig. 4, which clearly showcases the receptive field remapping.

We have included these new results in revised Figs. 3a, 4c–e and new Fig. 8, along with the following discussions:

Page 17: *“More specifically, experimentally results have demonstrated that the slope is increased upon application of a higher modulatory input (Fig. 4d), and hence the gain modulation has a multiplicative nature¹⁷.”*

Page 27: *“The gain modulation of the spiking neuron (Fig. 4) in the present work can also enrich the functionality of memristive neural networks. In biology, gain modulation is a canonical neural computation involved in many brain functions¹⁷ and plays a key role in maintaining visual stability when human eyes make frequent saccadic movements⁵⁰. Such stability is achieved by temporal expansion of receptive*

fields (RFs) of the neurons encoding object locations, named RF remapping, to compensate the shift of retinal locations of objects caused by saccade, such that the neural system will not perceive abrupt visual changes during the saccade. The above RF remapping is enabled by gain modulation, where an efference copy of the saccade serves as a gain signal that in turn triggers temporal propagation of the neuronal responses between current RF (CRF) and future RF (FRF)⁵⁰. The abovementioned receptive field remapping could significantly enhance the stability of artificial visual systems if it can be implemented in hardware, since similar instability issue can arise in real world, for example image shifts due to camera shaking may be confounded with real movements of objects.

Based upon the memristive neurons with gain modulation (Fig. 4), a one-dimensional network model is constructed, which includes 200 neurons connected in an uni-direction, as shown in Fig. 8a. Denote d_i as the input to neuron i , whose dynamics is given by:

$$\tau \frac{dd_i}{dt} = -d_i + W_{i,i-1} [f(d_{i-1}, m(t)) - \theta_{thr}]_+ + I_{ext,i} \quad (5)$$

where τ is the time constant, $W_{i,i-1}$ is the connection strength from neuron $i-1$ to i , $I_{ext,i}$ is the visual input, and θ_{thr} is a threshold. The symbol $[x]_+ = x$ for $x > 0$, and otherwise $[x]_+ = 0$. Only above θ_{thr} , i.e. $f(d_i, m) > \theta_{thr}$, the neuronal response can propagate to the neighborhood. By this, the modulation signal m controls the propagation of network activity. The firing rate of the neuron is given by Eq. (2), and the input modulation takes the multiplicative form as shown in Fig. 4.

The simulated network clearly shows that a visual input at FRF can only evoke a weak response of the neuron without modulatory signals (Fig. 8d). In stark contrast, when a modulatory signal is applied, it triggers a propagation of neuronal responses from FRF to CRF, as if the receptive field of the neuron is temporally expanded (Fig. 8b), which disappears when the modulatory signal terminates. Such result therefore demonstrates that memristive networks consisting of the spiking neurons with gain modulation can achieve RF remapping for enhanced visual stability, implying the great

potential of memristive neural networks in neuromorphic computing.”

Figure 4. Neuronal gain modulation of NbO_x based neuron. (a) Schematic diagram of the rate-coded neuronal signaling. The driving input and modulatory input affect output firing rate jointly. (b) Schematic diagram of neuronal gain modulation by modulatory inputs, including input modulation (orange line) and output modulation (purple line). The blue line is the I - O curve with only the driving input. (c) Neuron responses triggered by only the driving input (1.2 V amplitude, 1 μ s width, 0.4 μ s interval) applied on S_1 (bottom), and triggered by the driving input (1.2 V amplitude) and modulatory input (1 μ s width, 0.4 μ s interval) with different amplitudes of 0.4 V, 0.5 V, 0.6 V and 0.7 V, applied simultaneously to nodes S_1 and S_2 , as shown in the middle and top of the panel. (d) Experimental neuron responses showing multiplicative gain modulation when applying different modulatory inputs ($m = 0, 0.4, 0.5, 0.6$ and 0.7 V). The background lines indicate the overall trend in data. (e) Fitting lines calculated by Eq. (2).

Figure 8. A network of memristive neurons with gain modulation for receptive field remapping.

(a) A one-dimensional network where neurons are connected in an uni-direction. Each neuron receives a modulatory signal (orange arrow). A visual input is applied at the future RF (FRF) of a neuron (the yellow one). (b) The visual input at FRF and the modulatory signal (top, red line) triggers a propagation of neural activity from FRF to CRF of the yellow neuron, as if the RF of the yellow neuron is temporally expanded when the modulatory signal is applied, achieving RF remapping (right panel). The distance of remapping is controlled by the duration of the modulatory signal. In the simulation, the visual input is applied during 0–300 ms, while the modulatory signal $m = 0.8$ is applied during 300–1000 ms. $\theta_{thr} = 0.41$, $W_{i, i-1} = 0.095$, $\tau = 10$ ms, $dt = 1$ ms, $I_{ext, 0} = 0.09$.

3. The load resistor (in Figure 2), from the reviewer’s point of view, should be the synaptic device and not part of the neuron. The reviewer suggests the authors provide a pulse count vs. input frequency plot so that it can be compared with those in Figure 4.

Our response: Thanks for the suggestion. We fully agree that the load resistor is indeed equivalent with the synaptic device. As a result, we have revised the schematic

in Fig. 2d to reflect the correct implication on this point, as also appended below. The following revised sentences were also added to Page 7 “*The TS behavior of the Pt/Ti/NbO_x/Pt/Ti device can mimic the dynamics of an ion channel located near the soma of a neuron, while the membrane capacitance is represented by C_m*”.

Figure 2. (d) Illustration of an ion channel embedded in cell membrane of a biological neuron, with corresponding circuit based on NbO_x devices for implementation of a spiking neuron.

We want to thank the reviewer for the suggestion of providing a pulse count vs. input frequency plot. We have performed additional experiments, as shown in Supplementary Fig. 7. The pulse count vs. input frequency plot is qualitatively consistent with the case without modulatory input ($m = 0$) in Fig. 4d.

To address the question, the new results are now included as Supplementary Fig. 7 in the revised manuscript, along with the following discussion in Page 9 of the main text: “*Supplementary Fig. 7 further exhibits a plot of spike count vs. input frequency (with R_L of 3.6 k Ω , C_m of 30 pF, pulse width of 1 μ s, and pulse amplitude of 1.2 V), where the pulse count increases as the input frequency increases and get saturated at very high frequency (see further discussion in Supplementary Note 2)*”.

Supplementary Figure 7. A pulse count vs. input frequency plot of the artificial neuron (with R_L of 3.6 k Ω , C_m of 30 pF, pulse width of 1 μ s, and pulse amplitude of 1.2 V).

4. The reviewer suggests using a circuit diagram (or microscopic images, etc.) in Figure 3 in the main text to make it clear about the physical connection. In addition, since the pulse count should be discrete values, while that in the 3D plot in Fig. 3c and 3f seem to be continuous values, could the authors comment on it?

Our response: We would like to thank the reviewer for the helpful suggestion. A circuit diagram is now included as Fig. 3a in the revised manuscript, along with the following sentences in Page 10 “*Therefore, we use the circuit diagram in Fig. 3a to emulate this function, where the load resistors (R_1 and R_2) in series with NbO_x threshold switching device serve as synapses distributed in space and a capacitor (C_m) is connected in parallel*”. Detailed experimental parameters for R_1 , R_2 and C_m in all the measurements shown in Figs. 3 and 4 are described in detail in Supplementary Table 1.

The reviewer is absolutely right that the pulse count should be discrete values. The 3D surface plots originally in Figs. 3c, f (now Fig. 3d, h in the revised figure) are the enveloping contours of the experimental data points. In order to avoid any further misleading, here we have revised Figs. 3c, f (now Fig. 3d, h) by highlighting the discrete experimental data points while fitting the data with by two empirical equations, according to the reviewer’s next suggestion in question #5, where the fitting results are shown as 3D surface plots. This is now clearly described in the figure caption, and the

revised Fig. 3 is appended below for the reviewer's convenience.

Figure 3. Spatiotemporal integration and dynamic logic in NbO_x based neuron. (a) The circuit diagram of the spiking neuron. (b) Schematic diagram of spatial summation with different input pulse amplitudes. (c) Neuronal response triggered by two input pulses (0.9 V amplitude, 1 μs width, interval 0.1 μs, 10 pulse cycles) applied on S_1 and S_2 individually and simultaneously. (d) The spatial summation at varied conditions and corresponding fitting results. The amplitudes of the two input spikes (V_1 and V_2) were systematically changed from 0.6 V to 0.9 V, and applied simultaneously to the respective nodes. (e) The input-output characteristics for “AND” logic with two input pulses applied on S_1 and S_2 . (f) Schematic diagram of spatial summation with different input pulse intervals. (g) Neuronal response triggered by two input pulses (0.8 V amplitude, 1 μs width, interval 0.1 μs, 10 pulse cycles) applied on S_1 and S_2 individually and simultaneously. (h) The spatial summation at varied conditions and corresponding fitting results. The intervals of the two input spikes were systematically changed from 0.05 μs to 0.4 μs. (i) The input-output characteristics for “XOR” logic with two input pulses applied on S_1 and S_2 . (j) Schematic diagram of spatiotemporal summation with different time intervals of input pulses. (k) Neuronal response triggered by two input pulses (0.9 V amplitude, 1 μs width, 0.1 μs interval) applied on S_1 and S_2 with different time intervals of 0, 2.2 and -2.2 μs. (l) Spatiotemporal summation results when the time interval of two input spikes is changed from -6.6 to 6.6 μs. The solid line is fitting result by Eq. (1).

5. The authors provide an empirical function for the spatiotemporal summation, which is not used in the neural network demonstration. Since the summation that is used in the later demos is the spatial summation, could the authors provide an equation for Fig. 3c and Fig. 3f?

Our response: Thanks for the very helpful suggestion. In light of this advice, we have fitted the experimental results in Figs. 3c, f (now Fig. 3d, h in the revised figure) by the following empirical equations, and the fitting results are shown as the 3D surface plots together with the discrete experimental data points in Fig. 3d, h.

For Fig. 3d:

$$f(V_1, V_2) = b_1 e^{\beta_1 (V_1 - 0.6)} + b_2 e^{\beta_2 (V_2 - 0.6)} + b_3 (V_1 - 0.6)(V_2 - 0.6) + b_4$$

where $b_1 = 37.7$, $b_2 = 47.41$, $b_3 = -221.3$, $b_4 = -85.11$, $\beta_1 = 1.335$, $\beta_2 = 1.09$

and for Fig. 3h:

$$f(\Delta t_1, \Delta t_2) = c_1 e^{-\lambda_1 \Delta t_1} + c_2 e^{-\lambda_2 \Delta t_2} + c_3 \Delta t_1 \Delta t_2 + c_4$$

where $c_1 = 15.11$, $c_2 = 10.47$, $c_3 = 45.74$, $c_4 = -10.58$, $\lambda_1 = 1.694$, $\lambda_2 = 1.737$

These new results have been included in the revised Fig. 3 (as can be found in the response to previous question) and discussed in Supplementary Note 3.

6. The demonstrated Boolean logic (AND, OR) is a neuron connected with two synapses (two load resistors) for a linear neural network, which is not entirely new. It could be more interesting to demonstrate that the nonlinear dynamic (spatiotemporal input?) of the artificial neuron plays a role in the network, e.g. to implement a logic complete NOR or NAND function.

Our response: We would like to greatly thank the reviewer for the very constructive comment. Drawing inspiration from this suggestion, we have used the spiking neuron in the present study to implement XOR function, which is a typical Boolean logic that

is not linearly separable and therefore cannot be realized by a linear neural network (Minsky, M. L. & Papert, S. A. Perceptrons, MIT Press, Cambridge, MA, USA, 1969). Specifically, the input voltages to S_1 or S_2 serve as the input variables p and q , respectively. 0 V is defined as logic “0” for the inputs, and the logic “1” for p and q are defined to be 0.9 V and -0.9 V, respectively. The firing frequency of the artificial neuron is taken as the logic output, whose threshold is set to 0.3 MHz. Given the fact that the threshold switching in NbO_x is independent on the input voltage polarity (Fig. 2b, Supplementary Figs. 4 and 5), the firing frequency of the neuron exceeds the threshold when either p or q is “1”, but cannot reach the threshold when $p = 1$ and $q = 1$ are applied simultaneously, as experimentally demonstrated in Fig. 3i and Supplementary Fig. 8. The successful implementation of logic functions that are not linearly separable, like XOR, further enhances the potential of the present artificial neuron in achieving complex computing.

We have added the new results in Fig. 3i and Supplementary Fig. 8, along with further discussion in Page 12 “*More importantly, the spiking neuron in the present study can also be utilized to implement XOR function, which is a typical Boolean logic that is not linearly separable and therefore cannot be realized by a linear neural network⁴⁰. To do this, the neuronal threshold is set to 0.3 MHz, and the logic “1” for S_1 or S_2 is designated to be 0.9 V and -0.9 V, respectively. Given the fact that the threshold switching in NbO_x is independent on the input voltage polarity (Fig. 2b, Supplementary Figs. 4 and 5), the firing frequency of the neuron exceeds the threshold when either p or q is “1” but cannot reach the threshold when $p = 1$ and $q = 1$ are applied simultaneously, as experimentally demonstrated in Fig. 3i and Supplementary Fig. 8. The successful implementation of logic functions that are not linearly separable, like XOR, further demonstrates the potential of the artificial neuron in achieving complex computing.*”

Figure 3. Spatiotemporal integration and dynamic logic in NbO_x based neuron. (a) The circuit diagram of the spiking neuron. (b) Schematic diagram of spatial summation with different input pulse amplitudes. (c) Neuronal response triggered by two input pulses (0.9 V amplitude, 1 μ s width, interval 0.1 μ s, 10 pulse cycles) applied on S_1 and S_2 individually and simultaneously. (d) The spatial summation at varied conditions and corresponding fitting results. The amplitudes of the two input spikes (V_1 and V_2) were systematically changed from 0.6 V to 0.9 V, and applied simultaneously to the respective nodes. (e) The input-output characteristics for “AND” logic with two input pulses applied on S_1 and S_2 . (f) Schematic diagram of spatial summation with different input pulse intervals. (g) Neuronal response triggered by two input pulses (0.8 V amplitude, 1 μ s width, interval 0.1 μ s, 10 pulse cycles) applied on S_1 and S_2 individually and simultaneously. (h) The spatial summation at varied conditions and corresponding fitting results. The intervals of the two input spikes were systematically changed from 0.05 μ s to 0.4 μ s. (i) The input-output characteristics for “XOR” logic with two input pulses applied on S_1 and S_2 . (j) Schematic diagram of spatiotemporal summation with different time intervals of input pulses. (k) Neuronal response triggered by two input pulses (0.9 V amplitude, 1 μ s width, 0.1 μ s interval) applied on S_1 and S_2 with different time intervals of 0, 2.2 and -2.2 μ s. (l) Spatiotemporal summation results when the time interval of two input spikes is changed from -6.6 to 6.6 μ s. The solid line is fitting result by Eq. (1).

Supplementary Figure 8. XOR logic implemented using the NbO_x neuron. 0 V is defined as logic “0” for the inputs, and the logic “1” for p and q are defined to be 0.9 V and –0.9 V, respectively. Output of the neuron when “ $p = 1, q = 0$ ” (top panel), “ $p = 0, q = 1$ ” (middle panel), and “ $p = 1, q = 1$ ” (bottom panel) demonstrates a XOR function.

7. The scaled MNIST simulation is based on the experimental pulse response of the TaO_x synapses, but the simulated neural network is trained offline. Could the authors comment on how the synaptic device dynamics are used in the inference operation? The voltage-spike number response of the NbO_x neuron seems to be different from that shown Fig. 2f. Could the authors comment on it?

Our response: Thanks for raising this question, which we didn’t explain clearly. After the simulated neural network is trained offline, the trained weights are mapped onto TaO_x crossbar array, where the experimental pulse responses of TaO_x synapses are used for the mapping process. However, the reviewer is absolutely right that this is more related to the *static* characteristics of the synapses instead of *dynamics*, and therefore we have performed new simulations on 3-layer spiking neural network which is trained *online*. These new results have been included in the revised version of Fig. 6, along with the following discussion in Page 21–23 of the revised manuscript:

“To show the potential of the present devices in the construction of large-scale fully memristive spiking neural networks (SNN), we performed a simulation based on

experimental data. Figure 6a illustrates a 3-layer spiking neural network, which is composed of 784 input neurons, 100 hidden neurons and 10 output neurons, where the 784 inputs and 10 outputs correspond to a MNIST data size of 28×28 and 10 possible classes (from 0 to 9), respectively. We evaluate the network performance by MNIST handwritten digit classification, and detailed simulation process is shown in Fig. 6b, where the memristive SNN is trained online by backpropagation based on experimentally measured electrical characteristics of TaO_x synapses (Supplementary Fig. 14a, Supplementary Note 7) and the NbO_x neuron (Supplementary Note 7). As the first step in forward pass, a 28×28 MNIST image is converted to a 1×784 vector, and each pixel value is used to generate a Poisson event, i.e. a random voltage spike. The spiking possibility is higher if the corresponding pixel value is larger. During the simulation lasting for 500 time steps, there is a 784-spike-event vector at every time step and a train of 500 spikes for each input neuron. These spike trains are then fed into the memristor crossbar and converted to weighted current sums through the columns. A row of transimpedance amplifier can be used to amplify and convert the currents to analog voltages of $(-2, 2 \text{ V})$. The neurons can then integrate the analog voltages and generate spikes when reaching the firing threshold, which propagate to the next layer for similar processes. At last, the spiking numbers of output neurons are counted, and the index of most frequently spiking neuron is taken as the prediction result. During the backward propagation of errors, since the input-output function of a spiking neuron is a step function, which has infinite gradient, it is replaced by a soft activation function, e.g. sigmoid in the present case, to get the gradient and then the synaptic weights are adjusted accordingly (see Supplementary Note 7 for more details). Such training is performed for 100 epochs (Supplementary Fig. 14c).

Figure 6c shows that the 10 output neurons learnt specific digits during the training process, and depicts also the statistics of the firing numbers issued by 10 neurons in the case of input pictures corresponding to the category numbers themselves, where the input picture is correctly identified in most cases. The 3-layer SNN reaches a training accuracy of 81.45% after 100 epochs (Supplementary Fig. 14c), and the classification

accuracy of the simulated network is 83.24% on MNIST test dataset. The inference latency can be as low as 10 time steps, which is 10 μ s, with each neuron representing prediction result firing ~ 3.5 spikes on average. Figure 6d further shows a confusion matrix of the testing results from the 10,000 MNIST test dataset. As a measure on the classification accuracy, the confusion matrix in Fig. 6d displays the classification result in each column while the expected (actual) result in each row, where the number of instances is depicted by the color bar. As a result, the confusion matrix allows direct visualization of the firing number distribution of the trained output neurons in response to the test inputs, demonstrating that the test inputs are well classified after training.”

Figure 6. A 3-layer spiking neural network by simulation. (a) Schematic of the spiking neural network for MNIST classification. Input images are first converted to Poisson spike trains, where the spiking rates are proportional to pixel values. The spike trains are then input to the SNN, weighted and integrated by the neurons. At last, we count the spiking rates of output neurons to get the prediction results. (b) Flow chart of the simulation process. In the forward phase (yellow arrow), input spike trains are weighted by

the memristor crossbar and then integrated on neurons, and the spiking rates of output neurons are used in loss computing. In the backward phase (purple arrow), a soft function like sigmoid is used as an alternative to neuron spike function (step function) in gradient computing. The computing units in green boxes are simulated based on experimental data, while the units in blue boxes are implemented by software. (c) In the case where the input picture is correctly identified, the statistics of the firing numbers issued by ten neurons in the case of input pictures corresponding to the MNIST numbers themselves. Statistics of firing numbers of the category neurons in response to different inputs, showing that the inputs are classified correctly in most cases. (d) Averaged confusion matrix of the testing results, showing that the test inputs are well classified after training.

The seemingly different neuronal responses in spike count vs. voltage amplitude between Supplementary Fig. 9b (now Supplementary Fig. 14b) and Fig. 2f lies in two aspects: circuit parameter setup and device-to-device variation. While the R_L used in Fig. 2f is 3.6 k Ω , the R_L used in Supplementary Fig. 14b is 2.2 k Ω , and the numbers of applied pulses in Fig. 2f and Supplementary Fig. 14b were 20 and 10, respectively, as noted already in Methods and the caption of Supplementary Fig. 9b (now Supplementary Fig. 14b). The lower load resistance decides that the NbO_x neuron can start to fire at a lower amplitude of ~0.95 V, and the spike count under the same voltage amplitude is generally higher in the latter case, especially considering the smaller number of applied pulses (Supplementary Fig. 14b). The remaining difference can be attributed to device-to-device variations. However, in both cases the spike count shows an overall increasing trend, as the pulse amplitude increases. We have added related discussion in Supplementary Note 7 to address this point.

8. Since the volatile memristor might age very quickly as each neuron fire event will switch it. Could the authors comment on endurance performance, and how does it affect the neural network performance? The other concerns are the cycle-to-cycle variation and device-to-device variation, and speed/power advantages over competitors, but they are less important in this study, given it is a proof-of-concept at the current stage.

Our response: We would like to thank the reviewer for raising these points. We have performed new experiments to measure the endurance of the device, and the results show that the device can still function correctly after $>10^9$ switching cycles, as shown in Supplementary Fig. 2. Such endurance is promising for applications, but the reviewer is absolutely right that the requirement for switching in every firing event places high demand for the aging property of the devices. Fortunately, recent studies revealed that the threshold switching effects may not be from insulator-metal transition as believed previously but can be interpreted by a trap-assisted conduction mechanism similar to Poole-Frenkel model with moderate Joule heating, which actually suggest an electronic nature and much lower switching temperature (Refs. R1–R2). This may indicate a potential for further optimization on the endurance of NbO_x based devices.

Supplementary Figure 2. Endurance of Pt/Ti/NbO_x/Pt/Ti threshold switching device. (a) I - V characteristics of Pt/Ti/NbO_x/Pt/Ti device after 1, 10^4 , 10^6 , 10^7 , 10^8 and 10^9 switching cycles. **(b)** The firing behavior of Pt/Ti/NbO_x/Pt/Ti device after 1, 10^4 , 10^6 , 10^7 , 10^8 and 10^9 switching cycles.

In light of the reviewer's advice, we have also examined the cycle-to-cycle and device-to-device variation of the devices. Supplementary Fig. 4a shows the I - V characteristics of the NbO_x device in 50 repeated cycles, showing excellent cycle-to-cycle uniformity. The cycle-to-cycle fluctuations in $V_{\text{th, pos}}$, $V_{\text{hold, pos}}$, $V_{\text{th, neg}}$, $V_{\text{hold, neg}}$ as well as high and low resistance states are further plotted in Supplementary Fig. 4b, c, which once again demonstrates very low cycle-to-cycle variation.

Supplementary Fig. 5a–c exhibits I - V characteristics measured in 10 different

Pt/Ti/NbO_x/Pt/Ti devices and the device-to-device distributions of $V_{th, pos}$, $V_{hold, pos}$, $V_{th, neg}$, $V_{hold, neg}$ as well as high and low resistance states, showing relatively low device-to-device variations. These results in Supplementary Figs. 4 and 5 demonstrate that the Pt/Ti/NbO_x/Pt/Ti devices have acceptable cycle-to-cycle and device-to-device variations, making them qualified for building artificial neurons.

Supplementary Figure 4. Cycle-to-cycle variation of the NbO_x threshold switching device. (a) I - V characteristics of the device in 50 repeated cycles. (b) Cumulative plots of $V_{th, pos}$, $V_{hold, pos}$, $V_{th, neg}$, $V_{hold, neg}$. (c) Distributions of high and low resistance states of the NbO_x device in 50 repeated cycles.

Supplementary Figure 5. Device-to-device variation of the NbO_x threshold switching device. (a) I - V characteristics of the device measured in 10 different Pt/Ti/NbO_x/Pt/Ti devices. (b) Distributions of $V_{th, pos}$, $V_{hold, pos}$, $V_{th, neg}$, $V_{hold, neg}$ in 10 NbO_x devices. (c) Distributions of high and low resistance states in 10 NbO_x devices.

We have further compared the present work with existing approaches for implementing artificial neurons in terms of the speed and power consumption, including neurons based on diffusive memristors (Refs. R3–R9), threshold switching devices including NbO_x (Refs. R10–R16) and VO_x (Refs. R17–R23), as well as phase change materials like Ge₂Sb₂Te₅ (Refs. R24–R26), along with our NbO_x devices in the

present work (Supplementary Table 2). Here, the power consumption refers to the peak power consumed by the threshold switching (or resistive switching) device when the neuron fires. It can be found that the switching speed of NbO_x threshold switching device in the present work is <50 ns from off- to on-state and <25 ns from on- to off-state (Supplementary Fig. 3). The intrinsic switching speed of NbO_x was reported to be <10 ns (Refs. R10–R13), so the switching speed of NbO_x based devices is very fast and promising. Previously reported results on the power consumption of NbO_x based neurons have shown significant variation, ranging from 10–1600 μW (R14–R16). It is worthwhile noting that latest studies on threshold switching effects in NbO_x based on trap-assisted Poole-Frenkel conduction with moderate Joule heating (Refs. R1–R2) suggests much lower switching temperature and once again implies large potential in further optimizing the power consumption of NbO_x based neurons.

Supplementary Figure 3. Transient switching response of the NbO_x threshold switching device. (a) Current waveform (orange curve) of the NbO_x device upon application of the voltage waveform (blue curve). (b) The switching speed is <50 ns from off- to on-state. (c) The switching speed is <25 ns from on- to off-state.

Supplementary Table 2. Comparison of switching speed and power consumption for artificial neurons based on different materials and approaches.

Material	Speed	Power consumption
Ag/Ag:SiO _x N _y , Ag/HfO ₂ , Ag/SiO ₂ diffusive memristors	<500 ns ^[R3-R6]	1.2–480 μW ^[R7-R9]
NbO _x	<10 ns ^[R10-R13]	10–1600 μW ^[R14-R16]
VO _x	<700 ps ^[R17-R20]	23.75–2400 μW ^[R21-R23]
Ge ₂ Sb ₂ Te ₅	<20 ns ^[R24-26]	~4.3 μW ^[R26]
NbO _x (This work)	<50/25 ns	~392 μW

In order to address this question, we have added these new results as Supplementary Figs. 2–5 and Supplementary Table 2. We also added the following discussion into the main text to reflect the correct implication:

Page 6: “Symmetric hysteresis loops were observed in both bias polarities (Fig. 2b), and the device can operate properly after $>10^9$ switching cycles (Supplementary Fig. 2, Supplementary Note 1), showing stable volatile threshold switching (TS) characteristics. Transient electrical measurements show that the switching speed of NbO_x threshold switching device in the present work is <50 ns from off- to on-state and <25 ns from on- to off-state (Supplementary Fig. 3, Supplementary Note 1), with acceptable cycle-to-cycle and device-to-device variations, making them qualified for building artificial neurons (Supplementary Fig. 4 and 5, Supplementary Note 1).”

Page 30: “Compared with existing approaches for building artificial neurons, the NbO_x based neurons exhibit fast speed and comparable power consumption (see detailed comparison in Supplementary Table 2, Supplementary Note 9). Furthermore, latest mechanistic insights into NbO_x have revealed that the threshold switching might be achieved with lower switching temperature and hence reduced power consumption^{30,31},

which implies significant room for further device optimization based on NbO_x.”

Comments from Reviewer #3

Overall Remarks: This manuscript presents an artificial neuron, based on a volatile NbO_x memristor nano-device, which is capable to exhibit a number of functionalities, including the all-or-nothing threshold-driven spiking, spatio-temporal integration, as well as dynamic logic, and gain modulation among different dendritic inputs. A monolithically-integrated 4x4 fully-memristive neural network, consisting of volatile NbO_x memristor-based neurons and nonvolatile TaO_x memristor-based synapses, is experimentally demonstrated, showing the capability to perform pattern recognition through online learning using a simplified δ -rule. In my opinion the paper deserves acceptance in its current form. However, in preparation for the camera ready version, the authors should follow the guidelines below.

Our response: We would like to sincerely thank the reviewer for pointing out the novelty and significance of our study, and for the very detailed and constructive suggestions this reviewer kindly made. In this revised manuscript, we have carefully considered all the advices, performed new experiments/simulations (revised Figs. 3, 4 & 6, new Figs. 7 & 8, new Supplementary Figs. 2, 3, 5, 6, 7, 8, 14c & 15) and revised the manuscript accordingly. Detailed changes and responses can be found below.

1. A seminal paper on the modelling and investigation of the nonlinear dynamics of a locally-active NbO_x-based memristor from Ascoli et al. is

A. Ascoli, S. Slesazeck, H. Mähne, R. Tetzlaff, and T. Mikolajick, “Nonlinear dynamics of a locally-active memristor,” IEEE Trans. Circuits Syst.–I: Reg. Papers, vol. 62, no. 4, pp. 1165–1175, Apr. 2015, DOI: 10.1109/TCSI.2015.2413152.

In this paper the mechanisms behind threshold switching behaviour and negative

differential resistance effects in a NbO_x memristor device were elucidated by applying concepts from the theory of local activity. Please give credit to this work in the bibliography.

Our response: We would like to thank the reviewer for pointing out this important reference, which is now included into the revised manuscript as Ref. 34.

2. The Mott phase change transition was part of an early conjecture to explain NDR effects in NbO_x-based resistance switching memories. Over the past few years another hypothesis is gaining more credit. A trap-assisted Frenkel-Poole-based conduction mechanism in combination with mild Joule heating effects could most probably lie at the origin of threshold switching phenomena in NbO_x memristors. This was initially proposed and demonstrated in the following paper by Slesazek et al.:

S. Slesazek, H. Mähne, H. Wylezich, A. Wachowiak, J. Radhakrishnan, A. Ascoli, R. Tetzlaff, and T. Mikolajick, “Physical model of threshold switching in NbO₂ based memristors”, Journal of the Royal Society of Chemistry, vol. 124, no. 5, pp. 102318-102322, 2015, DOI: 10.1039/C5RA19300A

and later confirmed by Gibson et al. in the paper

Gibson, G. A. et al. An accurate locally active memristor model for S-type negative differential resistance in NbO_x. Appl. Phys. Lett. 108, 023505 (2016).

It is recommended to give appropriate credit to these two papers in the bibliography of your manuscript.

Our response: We would like to sincerely thank the reviewer for pointing out these two important references, which represent latest understandings on the threshold switching and NDR effects in NbO_x memristors. We have included these two works as Refs. 30 and 31, and revised the discussion on the threshold switching mechanism in Page 6 to reflect the correct implication “*Such threshold switching characteristics in NbO_x-based memristors have attracted extensive attention³⁰⁻³⁴, and recent studies*

revealed that such effects can be well interpreted by a trap-assisted conduction mechanism similar to Poole-Frenkel model with moderate Joule heating, therefore suggesting an electronic nature and much lower switching temperature than the previous insulator-metal transition model^{30,31}.”

3. In Fig. 1 the text “Pest-neuron” seems to be incorrect.

Our response: Thank you very much for pointing out this spelling mistake, which should be “Pre-neuron”. This has been corrected in the revised Fig. 1, as appended below.

Figure 1. Comparison of biological and artificial neurons. Schematic of biological neurons and synapses (left) compared with an artificial neural network (right).

4. Check the English throughout the manuscript. There are several typos.

Our response: Thanks for the suggestion. We have thoroughly checked the revised manuscript to remove any spelling mistakes.

5. In commenting Fig. 2(e) state that R_{OFF} is the initial resistance of the threshold switching device.

Our response: We would like to thank the reviewer for the remark. We have now

added definition for R_{OFF} in Page 8 of the revised manuscript to reflect this point: “ $R_{OFF} > R_L$, where R_{OFF} is the initial resistance of NbO_x threshold switching device”.

6. The voltage depicted in Fig. 2(e) is the pulse voltage applied to the left of R_L in Fig. 2(d) right? It would be nice if you could also plot the voltage across the threshold switching device.

Our response: We would like to thank the reviewer for the valuable remark. In light of this advice, we have performed new measurements on the voltage across the threshold switching device, and the results are included as Supplementary Fig. 6 in the revised manuscript, along with the following discussion in Page 8 of the main text: “*Supplementary Fig. 6 further shows the input voltage, current response and the voltage across the threshold switching device as functions of time, which illustrate the dynamic switching process clearly*”.

Supplementary Figure 6. The input voltage (blue curve), output current (orange curve) and the voltage across the threshold switching device (green curve) using the circuit in Fig. 2d.

7. In Fig. 2(d) the symbol used for R_L is typically adopted for a memristor. Please use the symbol of a resistor instead.

Our response: Thanks for the kind suggestion. Figure 2d has been re-drawn to use the right symbol for a load resistor. The revised figure is also appended below for the reviewer's convenience.

Figure 2. LIF neuron characteristics of NbO_x volatile memristors. (a) Schematic diagram of the threshold switching memristive device, which consists of a NbO_x layer between two Pt/Ti electrodes. (b) *I-V* characteristics of the device with 10 cycles. (c) Schematic illustration of the threshold switching in NbO_x devices. (d) Illustration of an ion channel embedded in the cell membrane of a biological neuron. The leaky integrate and fire (LIF) circuit with the NbO_x device is proposed to mimic the biological membrane. (e) Characterization of the LIF neuron under a continuous pulse train and the influence of varying capacitance (C_m) and resistance (R_L). (f) The firing response of the LIF neuron under different input and circuit conditions (See Methods). Changing the input parameters such as the pulse width, pulse interval and circuit parameters such as C_m and R_L , the firing pulse counts have an obvious change with

20 input pulse cycles.

8. In Fig. 2(e) replace R with R_L .

Our response: We would like to thank the reviewer for the careful review. This has been corrected, as can be found in revised Fig. 2 in response to question #7.

9. Caption of Fig. 2: replace R with R_L .

Our response: We would like to thank the reviewer for the careful review. This has been corrected, as can be found in revised Fig. 2 in response to question #7.

10. Caption of Fig. 3: “were systematically changed from 0.6 V to 0.9 V” -> “were systematically changed from 0.6 V to 0.9 V, and applied simultaneously to the respective nodes”

Our response: We would like to thank the reviewer for the kind suggestion. This sentence has been revised accordingly.

11. page 14 “significant” -> “significance”

Our response: We would like to thank the reviewer for the careful review. This spelling mistake has been corrected in the revised manuscript.

12. The difference between input modulation and output modulation should be clarified.

Our response: In light of the advice, we have added the following discussion in Page 15 of the revised manuscript: *“There are also two forms of neuronal gain modulation through synaptic input. One is input modulation (orange line in Fig. 4b), where the rate coded input–output (I – O) relationship is shifted along the x -axis upon altering the modulatory input, and the other is output modulation (purple line in Fig. 4b), where I – O relationship is shifted along the y -axis upon altering the modulatory input^{17, 36}. For input modulation, the maximum value of the I – O relationship does not change with modulation, but for output modulation, the maximum transmission rate increases or decreases proportionally (Fig. 4b).”*

13. What does m signify? Which property of the modulatory input voltage V_m does it symbolise? Does it represent its amplitude V_m ?

Our response: Yes “ m ” represents the amplitude of the modulatory input. In order to avoid confusions, we have added its definition in Page 16 “... where m is the amplitude of the modulatory input”.

14. “ d_{50} is the value of the driving input...” -> “ d_{50} is the frequency of the driving input...”

Our response: We would like to thank the reviewer for the careful review. This sentence has been revised accordingly.

15. Why is the gain $\frac{\partial f}{\partial d}$ at $V_m=0.7$ V the same as at $V_m=0.6$ V? If m coincides with the amplitude V_m of the modulatory input voltage V_m , as I conjectured in comment 13), from eq. (4) one expects that the gain would increase with V_m .

Our response: Thanks for raising the question. The gain should indeed increase as V_m increases, as predicted from Eq. (4). However, this is not obvious in previous Fig. 4d given the very similar amplitudes of modulatory inputs (0.6 V vs. 0.7 V). We have therefore performed additional experiments using a series of modulatory inputs (0, 0.4, 0.5, 0.6, 0.7 V) to improve Fig. 4, in order to examine the nature of the neuronal gain more clearly. The experimental results unambiguously demonstrate that the neuronal gain (Fig. 4d), i.e. the slope of f - d curve increases as the amplitude of modulatory inputs increases, and hence the gain modulation has a multiplicative nature (Ref. 17). The revised Fig. 4 is now included in the revised manuscript, which is also appended below for the reviewer’s convenience. In addition, we have revised the model in Eqs. (2–4) slightly to better fit the new experimental data (Fig. 4e), which has the same input gain modulation property as described in the previous model. We have included the following discussion in the revised manuscript to address this point:

Page 17: “*More specifically, experimentally results have demonstrated that the slope is increased upon application of a higher modulatory input (Fig. 4d), and hence the*

gain modulation has a multiplicative nature¹⁷.”

Figure 4. Neuronal gain modulation of NbO_x based neuron. (a) Schematic diagram of the rate-coded neuronal signaling. The driving input and modulatory input affect output firing rate jointly. (b) Schematic diagram of neuronal gain modulation by modulatory inputs, including input modulation (orange line) and output modulation (purple line). The blue line is the *I-O* curve with only the driving input. (c) Neuron responses triggered by only the driving input (1.2 V amplitude, 1 μs width, 0.4 μs interval) applied on *S*₁ (bottom), and triggered by the driving input (1.2 V amplitude) and modulatory input (1 μs width, 0.4 μs interval) with different amplitudes of 0.4 V, 0.5 V, 0.6 V and 0.7 V, applied simultaneously to nodes *S*₁ and *S*₂, as shown in the middle and top of the panel. (d) Experimental neuron responses showing multiplicative gain modulation when applying different modulatory inputs (*m* = 0, 0.4, 0.5, 0.6 and 0.7 V). The background lines indicate the overall trend in data. (e) Fitting lines calculated by Eq. (2).

16. Last line at page 16: “1.2 V amplitude and” -> “1.2 V amplitude) and”

Our response: We would like to thank the reviewer for carefully reviewing our manuscript. This has been corrected in the revised manuscript.

17. First line at page 17: “0.7 V simultaneously are shown” -> “0.7 V, applied simultaneously to nodes *S*₁ and *S*₂, are shown”

Our response: We would like to thank the reviewer for the careful review. This sentence has been corrected in the revised manuscript.

18. Page 18 “the neurons have been successfully trained, and the neural network recognizes “1010”, indicating” → “the neuron has been successfully trained, since it recognizes “1010”, indicating”

Our response: We would like to thank the reviewer for the careful review. This sentence has been corrected in the revised manuscript.

19. Page 20. Where it says “The images in the test dataset are converted into analog voltage vectors and input to the memristor array”, were the analogue voltage vectors encoded in the form of spikes before being inserted as inputs to the memristive array? Also, which memristive array is this text referring to? Is it a NbO_x memristor array used to form the 784 neurons at the input of the perceptron network? Also, were the TaO_x memristors used to implement the weights of the perceptron network?

Our response: Thanks for raising the questions. In previous simulations, the image pixels with a gray scale were encoded with analog voltage vectors with different amplitudes in the range of 0-0.5 V and fixed pulse width of 3 μs, while the memristive array refers to the TaO_x memristor array used to implement the weights of the perceptron network and the NbO_x memristors were used to implement the output neurons.

In light of the reviewer’s question, we have performed new simulations on 3-layer spiking neural network, where the image pixels were encoded in spikes and the network is trained online. These new results have been included in the revised version of Fig. 6, along with the following discussion in Page 21–23 of the revised manuscript:

“To show the potential of the present devices in the construction of large-scale fully memristive spiking neural networks (SNN), we performed a simulation based on experimental data. Figure 6a illustrates a 3-layer spiking neural network, which is

composed of 784 input neurons, 100 hidden neurons and 10 output neurons, where the 784 inputs and 10 outputs correspond to a MNIST data size of 28×28 and 10 possible classes (from 0 to 9), respectively. We evaluate the network performance by MNIST handwritten digit classification, and detailed simulation process is shown in Fig. 6b, where the memristive SNN is trained online by backpropagation based on experimentally measured electrical characteristics of TaO_x synapses (Supplementary Fig. 14a, Supplementary Note 7) and the NbO_x neuron (Supplementary Note 7). As the first step in forward pass, a 28×28 MNIST image is converted to a 1×784 vector, and each pixel value is used to generate a Poisson event, i.e. a random voltage spike. The spiking possibility is higher if the corresponding pixel value is larger. During the simulation lasting for 500 time steps, there is a 784-spike-event vector at every time step and a train of 500 spikes for each input neuron. These spike trains are then fed into the memristor crossbar and converted to weighted current sums through the columns. A row of transimpedance amplifier can be used to amplify and convert the currents to analog voltages of $(-2, 2 \text{ V})$. The neurons can then integrate the analog voltages and generate spikes when reaching the firing threshold, which propagate to the next layer for similar processes. At last, the spiking numbers of output neurons are counted, and the index of most frequently spiking neuron is taken as the prediction result. During the backward propagation of errors, since the input-output function of a spiking neuron is a step function, which has infinite gradient, it is replaced by a soft activation function, e.g. sigmoid in the present case, to get the gradient and then the synaptic weights are adjusted accordingly (see Supplementary Note 7 for more details). Such training is performed for 100 epochs (Supplementary Fig. 14c).

Figure 6c shows that the 10 output neurons learnt specific digits during the training process, and depicts also the statistics of the firing numbers issued by 10 neurons in the case of input pictures corresponding to the category numbers themselves, where the input picture is correctly identified in most cases. The 3-layer SNN reaches a training accuracy of 81.45% after 100 epochs (Supplementary Fig. 14c), and the classification accuracy of the simulated network is 83.24% on MNIST test dataset. The inference

latency can be as low as 10 time steps, which is 10 μ s, with each neuron representing prediction result firing ~ 3.5 spikes on average. Figure 6d further shows a confusion matrix of the testing results from the 10,000 MNIST test dataset. As a measure on the classification accuracy, the confusion matrix in Fig. 6d displays the classification result in each column while the expected (actual) result in each row, where the number of instances is depicted by the color bar. As a result, the confusion matrix allows direct visualization of the firing number distribution of the trained output neurons in response to the test inputs, demonstrating that the test inputs are well classified after training.”

Figure 6. A 3-layer spiking neural network by simulation. (a) Schematic of the spiking neural network for MNIST classification. Input images are first converted to Poisson spike trains, where the spiking rates are proportional to pixel values. The spike trains are then input to the SNN, weighted and integrated by the neurons. At last, we count the spiking rates of output neurons to get the prediction results. (b) Flow chart of the simulation process. In the forward phase (yellow arrow), input spike trains are weighted by the memristor crossbar and then integrated on neurons, and the spiking rates of output neurons are used

in loss computing. In the backward phase (purple arrow), a soft function like sigmoid is used as an alternative to neuron spike function (step function) in gradient computing. The computing units in green boxes are simulated based on experimental data, while the units in blue boxes are implemented by software. (c) In the case where the input picture is correctly identified, the statistics of the firing numbers issued by ten neurons in the case of input pictures corresponding to the MNIST numbers themselves. Statistics of firing numbers of the category neurons in response to different inputs, showing that the inputs are classified correctly in most cases. (d) Averaged confusion matrix of the testing results, showing that the test inputs are well classified after training.

20. End of page 20. “process, and the statistics of the firing numbers” -> “process, and depicts also the statistics of the firing numbers”

Our response: We would like to thank the reviewer for the careful review. This grammar mistake has been corrected in the revised manuscript.

21. The flow chart of the offline training in Fig. 6(b) could be described in some detail in the text.

Our response: Thanks for the suggestion. We have added the following discussion in Page 21–22 of the revised manuscript to explain the flow chart in Fig. 6(b): *“As the first step in forward pass, a 28×28 MNIST image is converted to a 1×784 vector, and each pixel value is used to generate a Poisson event, i.e. a random voltage spike. The spiking possibility is higher if the corresponding pixel value is larger. During the simulation lasting for 500 time steps, there is a 784-spike-event vector at every time step and a train of 500 spikes for each input neuron. These spike trains are then fed into the memristor crossbar and converted to weighted current sums through the columns. A row of transimpedance amplifier can be used to amplify and convert the currents to analog voltages of $(-2, 2 \text{ V})$. The neurons can then integrate the analog voltages and generate spikes when reaching the firing threshold, which propagate to the next layer for similar processes. At last, the spiking numbers of output neurons are counted, and the index of most frequently spiking neuron is taken as the prediction result. During the backward propagation of errors, since the input-output function of a spiking neuron is*

a step function, which has infinite gradient, it is replaced by a soft activation function, e.g. sigmoid in the present case, to get the gradient and then the synaptic weights are adjusted accordingly (see Supplementary Note 7 for more details). Such training is performed for 100 epochs (Supplementary Fig. 14c)."

22. The definition of the confusion matrix could be provided in the text.

Our response: We would like to thank the reviewer for the advice. In light of this, we have added further definition of the confusion matrix in Page 22 of the revised manuscript: *"As a measure on the classification accuracy, the confusion matrix in Fig. 6d displays the classification result in each column while the expected (actual) result in each row, where the number of instances is depicted by the color bar. As a result, the confusion matrix allows direct visualization of the firing number distribution of the trained output neurons in response to the test inputs, demonstrating that the test inputs are well classified after training."*

23. The paper provides no proof of evidence for the emergence of a Mott transition in the NbO_x memristor nano-device over the course of the threshold switching process. Therefore I would omit the reference to such a transition in the discussion section.

Our response: The reviewer is absolutely right on this. We have removed the references to "Mott transition" from the Discussion section and other places of the revised manuscript, in order to reflect the correct indication on this point.

24. I have no additional comment regarding the Supplementary Information document.

Our response: Thanks for all the constructive comments above.

References

- R1. Slesazeck, S. et al. Physical model of threshold switching in NbO₂ based memristors. *RSC Adv.* **5**, 102318-102322 (2015).
- R2. Gibson, G. A. et al. An accurate locally active memristor model for S-type negative differential resistance in NbO_x. *Appl. Phys. Lett.* **108**, 023505 (2016).
- R3. Yoo, J., Park, J., Song, J., Lim, S. & Hwang, H. Field-induced nucleation in threshold switching characteristics of electrochemical metallization devices. *Appl. Phys. Lett.* **111**, 063109 (2017).
- R4. Grisafe, B. et al. Performance Enhancement of Ag/HfO₂ Metal Ion Threshold Switch Cross-Point Selectors. *IEEE Electron Device Lett.* **40**, 1602-1605 (2019).
- R5. Midya, R. et al. Anatomy of Ag/HfO₂-based selectors with 10¹⁰ nonlinearity. *Adv. Mater.* **29**, 1604457 (2017).
- R6. Wang, Z. et al. Memristors with diffusive dynamics as synaptic emulators for neuromorphic computing. *Nat. Mater.* **16**, 101-108 (2017).
- R7. Wang, Z. et al. Fully memristive neural networks for pattern classification with unsupervised learning. *Nat. Electron.* **1**, 137-145 (2018).
- R8. Lee, D. et al. Various threshold switching devices for integrate and fire neuron applications. *Adv. Electron. Mater.* **5**, 1800866 (2019)
- R9. Zhang, X. et al. An artificial neuron based on a threshold switching memristor. *IEEE Electron Device Lett.* **39**, 308-311 (2018).
- R10. Park, J. et al. NbO₂ based threshold switch device with high operating temperature (> 85 C) for steep-slope MOSFET (2mV/dec) with ultra-low voltage operation and improved delay time. *2017 IEEE Int. Electron Devices Meet. (IEDM)* **23**, 7. 1-7. 4 (2017).
- R11. Pickett, M. D. & Williams, R. S. Sub-100 fJ and sub-nanosecond thermally driven

- threshold switching in niobium oxide crosspoint nanodevices. *Nanotechnology* **23**, 215202 (2012).
- R12. Luo, Q. et al. Nb_{1-x}O₂ based Universal Selector with Ultra-high Endurance (> 10¹²), high speed (10 ns) and Excellent V_{th} Stability. *2019 Symposium on VLSI Technology*. T236-T237 (2019).
- R13. Wang, Z., Kumar, S., Nishi, Y. & Wong, H. S. P. Transient dynamics of NbO_x threshold switches explained by Poole-Frenkel based thermal feedback mechanism. *Appl. Phys. Lett.* **112**, 193503 (2018).
- R14. Pickett, M. D., Medeiros-Ribeiro, G. & Williams, R. S. A scalable neuristor built with Mott memristors. *Nat. Mater.* **12**, 114-117 (2013).
- R15. Moon, K. et al. High density neuromorphic system with Mo/Pr_{0.7}Ca_{0.3}MnO₃ synapse and NbO₂ IMT oscillator neuron. *2017 IEEE Int. Electron Devices Meet. (IEDM)* **17**, 6. 1-6. 4 (2015).
- R16. Jerry, M. et al. Ultra-low power probabilistic IMT neurons for stochastic sampling machines. *2017 Symposium on VLSI Technology*. T186-T187 (2017)
- R17. Becker, M. F. et al. Femtosecond laser excitation of the semiconductor-metal phase transition in VO₂. *Appl. Phys. Lett.* **65**, 1507-1509 (1994).
- R18. Cavalleri, A. et al. Femtosecond structural dynamics in VO₂ during an ultrafast solid-solid phase transition. *Phys. Rev. Lett.* **87**, 237401 (2001).
- R19. Jerry, M. et al. Dynamics of electrically driven sub-nanosecond switching in vanadium dioxide. *2016 IEEE Silicon Nanoelectronics Workshop (SNW)*. 26-27 (2016)
- R20. Yi, W. et al. Biological plausibility and stochasticity in scalable VO₂ active memristor neurons. *Nat. Commun.* **9**, 1-10 (2018).
- R21. Lin, J. et al. Low-voltage artificial neuron using feedback engineered insulator-to-metal-transition devices. *2016 IEEE Int. Electron Devices Meet. (IEDM)* **34**, 5.1-

5.4 (2016)

- R22. Shukla, N. et al. Ultra low power coupled oscillator arrays for computer vision applications. *2016 IEEE Symposium on VLSI Technology*. 1-2 (2016)
- R23. Jerry, M. et al. Phase transition oxide neuron for spiking neural networks. *2016 74th Annual Device Research Conference (DRC)* 1-2 (2016).
- R24. Loke, D. et al. *Science* **336**, 1566 (2012).
- R25. Cheng, H. Y. et al. Atomic-level engineering of phase change material for novel fast-switching and high-endurance PCM for storage class memory application. *2013 IEEE Int. Electron Devices Meet. (IEDM)* **30**, 6.1-6.4 (2013).
- R26. Tuma, T., Pantazi, A., Le Gallo, M., Sebastian, A. & Eleftheriou, E. Stochastic phase-change neurons. *Nat. Nanotechnol.* **11**, 693 (2016).

REVIEWERS' COMMENTS:

Reviewer #2 (Remarks to the Author):

The authors clearly addressed all my concerns in the revised manuscript with new experiment/simulation data. The new data successfully established the connection between the single neuron characteristics, especially the nonlinear dynamics, and the possible applications. The endurance performance of the volatile memristor is still concerning, but this work clearly shows the significant power of this method, after the device performance gets mature enough in the future. Therefore, the reviewer believes it is a significant step for neuromorphic computing and would be happy to support this work for publication.